# The assembly of neutrophil inflammasomes during COVID-19 is mediated by type I interferons

Luz E. Cabrera[1]*, Suvi T. Jokiranta[2,3], Sanna Mäki[1], Simo Miettinen[1,4], Ravi Kant[1,4,5], Lauri Kareinen[1,4], Tarja Sironen[1,4], Jukka-Pekka Pietilä[6,7], Anu Kantele[6,7], Eliisa Kekäläinen[2,3,8], Hanna Lindgren[9], Pirkko Mattila[9], Anja Kipar[4,10,11], Olli Vapalahti[1,4,8], Tomas Strandin[1]

1 Viral Zoonosis Research Unit, Medicum, Department of Virology, University of Helsinki, Helsinki, Finland, 2 Department of Bacteriology and Immunology, University of Helsinki, Helsinki, Finland, 3 Translational Immunology Research Program, Faculty of Medicine, University of Helsinki, Helsinki, Finland, 4 Department of Veterinary Biosciences, University of Helsinki, Helsinki, Finland, 5 Department of Tropical Parasitology, Institute of Maritime and Tropical Medicine, Medical University of Gdansk, Gdynia, Poland, 6 Human Microbiome Research Program, Faculty of Medicine, University of Helsinki, Helsinki, Finland, 7 Meilahti Vaccine Research Center MeVac, Department of Infectious Diseases, Inflammation Center, Helsinki University Hospital and University of Helsinki, Helsinki, Finland, 8 Division of Virology and Immunology, HUSLAB Clinical Microbiology, HUS Diagnostic Center, Helsinki University Hospital, Helsinki, Finland, 9 Institute for Molecular Medicine Finland (FIMM), HiLIFE, University of Helsinki, Helsinki, Finland, 10 Laboratory for Animal Model Pathology, Institute of Veterinary Pathology, Vetsuisse Faculty, University of Zurich, Zurich, Switzerland, 11 Department of Infection Biology & Microbiomes, Institute of Infection, Veterinary and Ecological Sciences, University of Liverpool, Liverpool, United Kingdom

* luz.cabreralara@helsinki.fi

**Data Availability Statement:** All relevant data are within the manuscript and its supporting

## Abstract

The severity of COVID-19 is linked to excessive inflammation. Neutrophils represent a critical arm of the innate immune response and are major mediators of inflammation, but their role in COVID-19 pathophysiology remains poorly understood. We conducted transcriptomic profiling of neutrophils obtained from patients with mild and severe COVID-19, as well as from SARS-CoV-2 infected mice, in comparison to non-infected healthy controls. In addition, we investigated the inflammasome formation potential in neutrophils from patients and mice upon SARS-CoV-2 infection. Transcriptomic analysis of polymorphonuclear cells (PMNs), consisting mainly of mature neutrophils, revealed a striking type I interferon (IFN-I) gene signature in severe COVID-19 patients, contrasting with mild COVID-19 and healthy controls. Notably, low-density granulocytes (LDGs) from severe COVID-19 patients exhibited an immature neutrophil phenotype and lacked this IFN-I signature. Moreover, PMNs from severe COVID-19 patients showed heightened nigericin-induced caspase1 activation, but reduced responsiveness to exogenous inflammasome priming. Furthermore, IFN-I emerged as a priming stimulus for neutrophil inflammasomes. These findings suggest a potential role for neutrophil inflammasomes in driving inflammation during severe COVID-19. Altogether, these findings open promising avenues for targeted therapeutic interventions to mitigate the pathological processes associated with the disease.

information files. RNA-seq data are deposited at the GEO (GSE272381 and GSE271808).

**Funding:** This work was financed by grants by the Academy of Finland to T.S. (321809), A.K. (336439 and 335527); grants by the Helsinki University Hospital funds to O.V. (TYH 2021343); EU Horizon 2020 programme VEO (874735) to O.V.; Finnish governmental subsidy for Health Science Research (TYH 2021315) to A.K.; Paulon Säätiö to L.E.C.; Suomen Lääketieteen Säätiö to L.E.C.; Jane and Aatos Erkko foundation to O.V. The funders had no role in study design, data collection and analysis, nor decision to publish, or preparation of the manuscript.

**Competing interests:** The authors have declared that no competing interests exist.

## Author summary

COVID-19, caused by the SARS-CoV-2, ranges from mild "flu"-like symptoms to severe respiratory distress or even death. Neutrophils are important cells of our immune system which are strongly involved in inflammatory responses, including those occurring in COVID-19. However, despite extensive research, the precise contribution of neutrophils to the pathogenesis of COVID-19 remains elusive, and further clarification on their role is still needed. In this study, we isolated neutrophils from COVID-19 patients and healthy controls to analyze changes in their gene expression profile and inflammatory functions. These analyses revealed a distinct type I interferon (IFN-I) gene signature expressed by mature, but not immature, neutrophils from severe COVID-19 patients, which was absent in mild cases and healthy controls. Additionally, neutrophils from severe COVID-19 showed signs of increased inflammasome activation, a protein complex that contributes to inflammation by releasing inflammatory cytokines. Notably, IFN-I alone was able to promote neutrophil inflammasome formation *in vitro* suggesting a direct link between IFN-I response and inflammasome formation during COVID-19. Furthermore, increased neutrophil inflammasome activity was detected also in a mouse model of COVID-19. These findings suggest a potential role for neutrophils in driving excessive inflammation during severe COVID-19, and a role for IFN-I in priming the assembly of inflammasomes in these cells.

## Introduction

Severe COVID-19 is characterized by a dysregulated immune response with an excessive production of pro-inflammatory cytokines and chemokines. Type I interferons (IFN-I) are critical antiviral cytokines in the innate immune responses against viral infections, drawing particular attention amidst the COVID-19 pandemic [1–3]. While the IFN-I response helps to limit virus replication [3], its prolonged and uncontrolled activation is detrimental to the overall health of the patient [4]. As part of the pro-inflammatory response, neutrophils are rapidly recruited to the site of infection in response to SARS-CoV-2 infection [5,6]. Prominent neutrophil recruitment in severe COVID-19 is associated with an increased number of immature low-density granulocytes (LDGs) in the circulation [7–9]. The increased production and subsequent early release of immature cells from the bone marrow occurs in response to emergency myelopoiesis [9]. This process is initiated by the body to enable the recruitment of innate immune cells into the tissues and to replenish the depleted leukocyte pool, in an effort to combat viral infections including SARS-CoV-2 [10]. However, the premature release of these cells could be associated with the increased degranulation and formation of neutrophil extracellular traps (NETs) reported during SARS-CoV-2 infection, to which LDGs have a higher propensity than polymorphonuclear cells (PMN) [5,6,11].

Neutrophils are involved in several aspects of inflammatory processes, including the release of reactive oxygen species (ROS) and other pro-inflammatory mediators such as Interleukin-6 (IL-6) and IL-8. In addition, recent reports on COVID-19 highlight that neutrophils could be a major source of inflammasome derived IL-1β, which has been implicated as a substantial contributor to COVID-19 pneumonia [12]. Inflammasomes are intracellular multiprotein complexes involved in the inflammatory response. In the presence of a pathogen, antigen recognition by the immune system triggers the assembly of the inflammasome, a step known as the first signal. This is followed by the recruitment of adaptor molecules that activate NOD-like receptor (NLR) family members and the binding of the apoptosis-associated speck-like protein (ASC), finally activating the inflammasome complex [13]. The triggered assembly of

this complex is known as the second signal. Studies have shown that SARS-CoV-2 infection induces significant inflammasome activation in circulating and lung-infiltrating myeloid cells, such as monocytes and neutrophils [14–17]. However, while the precise mechanism by which inflammasomes are activated in monocytes/macrophages is well established, less is known about molecular mechanisms of inflammasome formation in neutrophils. Thus, this study investigates the inflammasome formation in neutrophils during COVID-19 in more detail, also focusing on the different developmental stages of these cells. In addition, a recently established COVID-19 mouse model served to further explore the role of IFN-I in neutrophil inflammasome assembly.

## Materials and methods

### Ethics statement

The study was approved by the Ethics Committee of the Hospital District of Helsinki and Uusimaa (HUS/853/2020, HUS/1238/2020). All volunteers gave a written informed consent, in accordance with the Declaration of Helsinki. For animal experiments, experimental procedures were approved by the Animal Experimental Board of Finland (license number ESAVI/28687/2020).

### Patient population

Adult clinical patients with confirmed COVID-19 (RT-PCR positive for SARS-CoV-2) at Helsinki University Hospital (HUH) (hospitalized: n = 34; outpatients: n = 8) were enrolled in the present study. Blood samples were collected during hospitalization for the severe COVID-19 group, and after confirmation of diagnosis for the mild COVID-19 outpatient group. Samples for RNA sequencing were collected in 2020 and representing infections by the original and early SARS-CoV-2 variants, whereas samples for *ex vivo* culture experiments were collected in 2021–2022 likely representing infections by omicron subvariants of SARS-CoV-2. As controls, healthy blood donors were included for RNA sequencing (n = 7, age 57 ± 7, male/female 3/4) and *ex vivo* culturing experiments (n = 9, age 38 ± 14, male/female 4/5). For clinical correlation analysis, severe COVID-19 patients were further categorized by severity based on their need for hospitalization and oxygen supplementation, as described previously [7]. For each patient, medical history and clinical data were collected through retrospective patient record review and are presented for the severe COVID-19, hospitalized patients in Table 1 and as previously described [7]. Calprotectin was measured from serum (diluted 1:1000) by ELISA, according to the manufacturer's protocol (calprotectin/S100A8 DuoSet kit, R&D systems).

The World Health Organization (WHO) Ordinal Scale for clinical improvement is a tool designed specifically to assess and measure the progression and clinical improvement of patients [67]. COVID-19 scoring: 1 = no limitations of activity, 2 = limitations of activity, 3 = no oxygen therapy, 4 = oxygen by mask or nasal cannulae, 5 = non-invasive ventilation or high-flow oxygen, 6 = invasive mechanical ventilation without other organ support, 7 = invasive mechanical ventilation with other organ support, 8 = dead. The baseline score represents the timepoint of the first laboratory sample taken, serving as a reference point for measuring improvement and establishing a starting point for comparison. In contrast, the worst score represents the most severe or critical state of the disease. None of the patients whose samples were used for RNA-seq underwent corticosteroid treatment.

### Isolation of granulocytes from human blood

Blood samples from COVID-19 patients and healthy controls (HC) were collected in EDTA vacutainer tubes and transported to the laboratory. Peripheral blood mononuclear cells

**Table 1. Clinical parameters of hospitalized patients (n = 34).**

| Parameter | Range (Mean + SD) or ratio |
|---|---|
| Age (years) | 22–80 (58.41 + 13.63) |
| Gender (male:female) | 23:11 |
| Hospitalization (days) | 3–38 (12 + 8.63) |
| WHO Ordinal scale for Clinical improvement: Baseline | 1–6 (4 + 1.07) |
| WHO Ordinal scale for Clinical improvement: Worst | 2–8 (4 + 1.42) |
| Deaths (count/total) | 1/34 |
| ICU (count/total) | 9/34 |
| Corticosteroid use (count/total) | 13/34 |
| Length of stay in ICU (days) | 1–27 (7.38 + 9.49) |
| Max CRP (mg/L) | 6–466 (159.56 + 104.06) |
| Min lymphocyte count ($E^9$/L) | 0.08–2.02 (0.79 + 0.42) |
| Max neutrophil count ($E^9$/L) | 1.08–29.86 (7.85 + 5.73) |
| Sampling day neutrophil count ($E^9$/L) | 0.64–20.52 (5.65 + 3.95) |

*Abbreviations: WHO = World Health Association. ICU = Intensive Care Unit. CRP = C reactive protein.*
*Min = minimum. Max = maximum.*

(PBMCs) or polymorphonuclear cells (PMNs) were isolated from whole blood by density gradient centrifugation using either Ficoll-Paque Plus (GE Healthcare) or Polymorphprep (Axis-Shield) respectively, following standard procedures including the use of 2 mM EDTA in PBS and red blood cell lysis with ACK lysis buffer (Lonza by Thermo Fisher). Subsequently, isolation of CD66[+] granulocytes (low-density granulocytes, LDGs) from the PBMC fraction was performed using the CD66abce MicroBead Kit (Miltenyi Biotec, Germany) with an MS column, according to the manufacturer's instructions. Both the positively selected CD66[+] LDGs and the isolated PMNs were then washed and counted, using a TC20 Automated Cell Counter (Bio-Rad Laboratories, Inc.) with trypan blue staining for dead cell exclusion. All described procedures in this section were done at room temperature. An aliquot of cells was lysed in Trizol reagent (Thermo Fisher Scientific, USA) and stored at –80˚C for later extraction of total RNA and subsequent RNA sequencing (RNA-seq) analysis.

## Caspase1 activity

Caspase1 activity was assessed in isolated cells after 2 h of culture (1 million cells/ml) using the caspase-Glo1 inflammasome assay (Promega) according to the manufacturer's protocol, with 2.5 μM nigericin (Invivogen) treatment as the activator. The resulting luminescence was measured by a Hidex Sense microplate reader (Hidex).

As another approach, caspase1 activity in isolated cells (1 million cells/ml) was measured by the fluorescent dye FAM-FLICA (Bio-Rad Laboratories). Cells were incubated with FAM-FLICA according to manufacturer's recommendations for 30 min in culture medium at 37˚C, after which cells were washed with PBS and analyzed by LSRII cytometer (BD Biosciences). Data was acquired with BD FACSDiva version 8.0.1 (BD Biosciences) software and further analysis was performed with the FlowJo software v10 (BD Biosciences).

## Soluble factor stimulation assays

Isolated granulocytes from HC and COVID-19 patients were cultured at 2 million cells/ml in RPMI 1640 supplemented with 10% fetal bovine serum (R10) at 37˚C. Cells were primed (1st

signal) with either LPS (20 ng/ml, Sigma Aldrich) or IFN-I (combination of $2.7*10^4$ IU/ml IFN-α and IFN-β, Immunotools) for 4 h, followed by activation ($2^{nd}$ signal) by 2.5 μM nigericin or monosodium urate crystals (MSU, 100 μg/ml, Invivogen) for an additional 4 h. For the 24 h stimulation experiments, nigericin was added to the cultured cells, in the presence or absence of inflammasome inhibitors MCC995 (2 μg/ml) and Ac-YVAD-FMK (20 μg/ml, both from Invivogen). Cells were pelleted by centrifugation at 400 G for 5 min and stored in Trizol at –80˚C for later RNA extraction whereas supernatants were used to measure IL-1β, IL-18, myeloperoxidase (MPO) and IL-8 by ELISAs according to the manufacturer's protocols (Duo-Set kits from R&D systems). LDH was measured in supernatants using Cyquant LDH cytotoxicity assay (ThermoFisher). Where indicated, priming and activation was performed in the presence of 100 μg/ml of anti-human IFNAR1 (Anifrolumab Biosimilar) or mouse IgG1 as control (both from Bio-X-Cell, New Hampshire, USA).

HL-60 cells (ATCC #CCL-240) were activated similarly to neutrophils after a 5-day differentiation period induced by 1% DMSO.

## Virus propagation

The SARS-CoV-2 hCoV-19/Finland/THL-202117309/2021 (delta strain B.1.617.2) and the mouse-adapted strain MaVie [18] were propagated in VeroE6-TMPRSS2 cells (kidney epithelial cells expressing the transmembrane protease serine 2) [19] grown in DMEM supplemented with 10% inactivated FCS, 100 IU/mL Penicillin, 100 μg/mL Streptomycin and 2 mM L-glutamine at 37˚C. The virus was purified from supernatants by ultracentrifugation (SW28 rotor, 27,000 rpm, 90 min, +4˚C) through a 0.22 μm-filtered 30% ultra-pure sucrose cushion (in PBS), to obtain virus preparations free of cell culture contaminants. Virus titers were calculated by the median tissue culture infectious dose (TCID50) after assessing cytopathic effects by crystal violet staining of cell cultures infected for 5 days with serially diluted virus.

## RNA sequencing

Neutrophils isolated from different cohorts comprised three PMN groups (severe COVID-19, mild COVID-19, and healthy controls), and one LDG group (given that these cells were rare in mild COVID-19 patients and HC, only LDGs from patients with severe COVID-19 were included).

cDNA synthesis from total RNA was performed according to Takara SMARTseq v4 Ultra-low input RNA kit for Sequencing user manual (Takara Bio, Mountain View, CA, USA) followed by Illumina Nextera XT Library preparation according to Illumina Nextera XT Reference Guide (Illumina, San Diego, CA, USA). UDI index setup was used for the Nextera XT libraries. Library quality check was performed using LabChip GX Touch HT High Sensitivity assay (PerkinElmer, USA) and libraries were pooled based on the concentrations acquired from the assay. The pooled libraries were quantified for sequencing using KAPA Library Quantification Kit (KAPA Biosystems, Wilmington, MA, USA) and sequenced on the Illumina NovaSeq6000 system for 200 cycles using S1 flow cell (Illumina, San Diego, CA, USA). Read length for the paired-end run was 2x101 bp. The human RNA-seq data are deposited at the GEO (accession number GSE#272381).

## RNA data analysis

Principal Component Analysis (PCA) and enrichment analyses were obtained using ExpressAnalyst [20]. Briefly, PCA was performed to identify patterns in the data and reduce the dimensionality of the dataset, where the top principal components were selected based on the percentage of variance explained. For enrichment analyses, Gene Set Enrichment Analysis

(GSEA) and Over-Representation Analysis (ORA) were performed on the top 5000 DE genes identified by DESeq2 (adjusted P value < 0.05, log2FC >1) [20]. GSEA was used to identify enriched signaling pathways using the Reactome database, while ORA was used to identify enriched pathways using the KEGG database. The resulting p-values were corrected for multiple testing using the Benjamini-Hochberg method, and pathways with a corrected p-value <0.05 were considered significant.

To visualize the expression patterns of the DE genes, the data was analyzed using the AltAnalyze software [21], which selected the top 118 genes based on correlation and determined the heatmap clustering, using the Euclidean distance metric and the complete linkage method. Then, the obtained heatmap was re-generated using heatmapper.ca [22] for better visualization.

CIBERSORTx, a machine learning algorithm that infers cell type proportions using a reference gene expression matrix of known cell types was used to perform RNA-seq deconvolution on the gene expression data to estimate the abundance of immune cell types in the samples [23]. The signature matrix used was taken from Lasalle *et al.* [8]. This reference matrix made use of a published whole-blood single-cell dataset [9], and included the main immune cell types: monocytes, NK cells, T lymphocytes, B lymphocytes, plasmablasts and neutrophils, the latter subclassified into mature and immature. The smaller subsets of granulocytes (eosinophils and basophils) are not considered separately and are most likely categorized as neutrophils in the bulk data deconvolution. Nonetheless, the resulting cell type proportions were used to compare the immune cell composition between groups.

Additionally, the determination of sample purity (>65% identified as neutrophils) served as a limiting parameter for the visualization of differentially expressed inflammasome related genes from the RNA sequencing results, which were selected and graphed in a heatmap using heatmapper.ca [22], clustered by complete linkage and ordered by Spearman's rank.

For the GSEA of the reanalyzed RNA-seq data from LaSalle *et al.* [8], we used the fgsea R package using MSigDB pathway sets, as specified in S1 Table.

## Volcano plots

To visualize differentially expressed (DE) genes between groups from human and mice RNA-seq results previously identified by DESeq2, a volcano plot was generated using GraphPad Prism. Genes with a P-adjusted value (padj or FDR) <0.05 were considered significant. Similarly, RNA sequencing data from GSE93996 [24] was reanalyzed, and all DE genes in ATRA-differentiated HL-60 cells were visualized in a volcano plot.

## Single cell transcriptomics data analysis

This study made use of the "COVID-19 Immune Atlas: integration of 5 public COVID-19 PBMC single-cell datasets" available online [25]. This standardized data collection contains cells from different assays (10x 3' v2, 10x 3' v3, 10x technology and Seq-Well) and consists of a total of 239,696 cells from the peripheral blood, 3,693 of which are neutrophils. These neutrophils were further subclassified as mature (59%) and immature (41%), based on the immune atlas predetermined cell classes. This was confirmed by a CD16b expression in mature neutrophils, and a higher CD66b expression in the immature population. This data was obtained from and analyzed in the Chan Zuckerberg CELLxGENE platform [25].

## Reverse transcription and quantitative PCR (RT-qPCR) for human selected human genes

Total RNA was extracted from unstimulated or *ex vivo* stimulated PMNs using the Trizol reagent (Invitrogen, USA) according to the manufacturer's protocol. Subsequently, cDNA

synthesis was performed using the RevertAid RT Reverse Transcription Kit (Thermo Scientific, USA) as per the manufacturer's instructions. Quantitative PCR (qPCR) was performed using the Stratagene model (Agilent Technologies) and SYBR Green/ROX master mix (Thermo Scientific, USA). The primer sequences for qPCR are presented in S2 Table.

Primer specificity was confirmed using melting curve analysis and dissociation curves. The relative expression levels of the genes of interest were calculated using the 2-ΔΔCT method and normalized to the expression of the housekeeping gene GAPDH. Baseline gene expressions of unstimulated samples were statistically assessed using the Mann-Whitney test, while the two-way ANOVA Tukey's multiple comparisons test was performed for the *ex vivo* stimulated samples.

## Mouse infections

Female BALB/c mice (Envigo, Indianapolis, IN, USA; 7 to 8 weeks, n = 36 in total) were transferred to the University of Helsinki biosafety level-3 (BSL-3) facility and acclimatized to individually ventilated biocontainment cages (ISOcage; Scanbur, Karl Sloanestran, Denmark) for 7 days with *ad libitum* water and food (rodent pellets). For subsequent experimental infection, the mice were placed under isoflurane anesthesia and inoculated intranasally with 50 μL of SARS-CoV-2 MaVie strain ($5*10^5$ TCID50/animal) or PBS (mock-infected control). Daily weighting of all mice was performed, and their well-being was carefully monitored for signs of illness (e.g., changes in posture or behavior, rough coat, apathy, ataxia). Euthanasia was performed by cervical dislocation under terminal isoflurane anesthesia. All animals were dissected immediately after euthanasia, and the lungs were sampled for multiple downstream analyses. The infections were performed as 4 separate experiments (exp): 1) Exp 1 included 8 mice infected with MaVie and 4 mock infected mice. At 2 days post infection (dpi), 4 infected and the mock infected mice were euthanized; the remaining infected mice were euthanized at 4 dpi. The right lung was sampled for virus-specific RT-qPCR (1/5) and neutrophil isolation (4/5), the left lung was fixed for histological and immunohistochemical examination. 2) Exp 2 included 8 infected and 4 mock infected mice of which half were euthanized at 2 dpi and 4 dpi, respectively. From these mice, both lung lobes were subjected to neutrophil isolation. 3) Exp 3 included 8 mice that were infected and immediately inoculated intraperitoneally with 250 μg of anti-mouse IFNAR-1 (n = 4) or mouse IgG1 control (n = 4) (Bio-X-Cell, New Hampshire, USA), and 4 mock-infected animals. All mice were euthanized at 2 dpi. Each 1/5 of the left lobe was processed for virus-specific RT-qPCR and histology/immunohistochemistry, respectively. The remaining 4/5 of the lungs served for neutrophil isolation. 4) Exp 4 included 4 animals in the following 5 groups: PBS-inoculated controls, infected animals euthanized at 2 dpi and at 4 dpi respectively, infected animals euthanized at 2 dpi with intraperitoneal injection of control IgG or anti-mouse IFNAR-1. Each 1/5 of the left lobe was processed for virus-specific RT-qPCR and histology/immunohistochemistry, respectively. The remaining lung tissue served to prepare single cell suspensions. Each 1 million cells were subjected to neutrophil quantification or caspase1 activity measurement by flow cytometry, the rest (approx. 8 million cells) to neutrophil isolation.

## Virus titration from mouse lungs

Supernatants of single cell suspensions from Exp 4 were used for virus titration by fluorescent focus forming unit (FFU)-based assay, in which Vero E6 cells were incubated with serially diluted supernatants for 24 hours at 37°C in growth medium. Cells were fixed with 4% PFA for 10 min, blocked and permeabilized with PBS containing 3% BSA and 0.2% TritonX-100 for 15 min and incubated with rabbit anti- SARS-CoV-2 receptor binding domain (RBD) [19]

for 1 hr, followed by anti-rabbit AlexaFluor488 conjugated secondary antibody (Thermo Scientific) for 1 hr. Fluorescence was observed with Zoe fluorescence imager (Bio-Rad laboratories) and the highest dilution not showing any RBD positive cells considered as the virus titer.

## Neutrophil quantification and caspase1 activity measurement from mouse lungs by flow cytometry

Single cell suspensions from mouse lungs of Exp 4 were subjected to flow cytometric quantification of Ly6G+ neutrophils. Cells were initially incubated with BV605-conjugated yellow live/dead dye (Thermo Scientific) for 15 min before addition of 1% FBS and a cocktail of antibodies recognizing CD3+ and CD19+ lymphocytes (FITC-conjugated clones 145-2C11 and 1D3, respectively, from Immunotools), Ly6G+ neutrophils (PE-Cy7-conjugated clone 1A8 from BD biosciences) and CD11b (APC-conjugated clone M1/70.15 from Immunotools). After incubation for 30 min at RT, cells were fixed with 2% paraformaldehyde for 30 min and washed with PBS. In parallel, single cell suspensions were stained with FAM-FLICA together with Ly6G antibody and 30 min incubations were performed in R10 at 37˚C before fixation. Finally, all cells were subjected to flow cytometric analysis with a three-laser (Blue/Red/Violet lasers) 14-color Fortessa LSRII cytometer (BD Biosciences). Data was acquired with BD FACSDiva version 8.0.1 (BD Biosciences) software and further analysis was performed with the FlowJo software v10 (BD Biosciences).

## Neutrophil isolation from mouse lungs

Neutrophil isolation was performed from the lungs of all mice. The dissected lung tissue was chopped into small pieces using scissors and enzymatically digested with a cocktail of Liberase (50 ug/ml; Roche #05401020001 from Merck) and DnaseI (100 ug/ml; Roche #11284932001 from Merck) in RPMI-1640 for 30 min at 37˚C. The resulting homogenate was diluted 10-fold in R10 and passed through a 70 μm Cell strainer (Pluriselect) to obtain a single-cell suspension. Neutrophils were isolated by positive selection using Ly6G-binding magnetic beads and MS columns according to the manufacturer's recommendations (Miltenyi Biotec). Neutrophils were isolated with a purity exceeding 95% based on flow cytometry analysis of Ly6G expression. Where indicated, isolated neutrophils were attached to glass slides through cytospin (800 g, 5 min), fixed with 4% PFA for 10 min and the nuclei stained with Hoechst33342.

## RNA sequencing of mouse neutrophils

Mouse neutrophils from Exp 1 were isolated, lysed in Trizol (Thermo Scientific) and the RNA extracted in the liquid phase using chloroform. RNA isolation was carried out using the Rneasy micro kit (Qiagen). Isolated RNA (1 ng) underwent whole transcriptome sequencing with ribodepletion. Briefly, RNA sequencing was performed using the Illumina Stranded with RiboZero library preparation method. Sample quality and integrity were assessed using TapeStation RNA analysis. Sequencing was conducted on the Illumina NextSeq platform, followed by standard bioinformatics analysis for gene expression quantification.

The service was provided by the Biomedicum Functional Genomics Unit at the Helsinki Institute of Life Science and Biocenter Finland at the University of Helsinki. The mouse RNA-seq data are deposited at the GEO (GSE271808).

## RT-qPCR of mouse samples

RNA was extracted from dissected lung samples (1/10 of the whole lung) as well as isolated neutrophils of mice in Exp 1, 3 and 4 using Trizol (Thermo Scientific) following the

manufacturers' instructions. The isolated RNA was directly subjected to one-step RT-qPCR analysis based on a previously described protocol using primer-probe sets detecting the viral genome encoding for the RNA-dependent RNA polymerase (RdRp [26], subgenomic E [27] as well as mouse OasL2, caspase1, IL1b and GAPDH (Applied biosystems #Mm01336189_m1, #Mm00438023_m1, #Mm00434228_m1 and #Mm99999915_g1 respectively, Thermo Scientific). The PCRs were performed with TaqPath 1-step master mix (Thermo Scientific) using AriaMx instrumentation (Agilent, Santa Clara, CA, USA).

## Histology and immunohistochemistry

From animals in Exp 1, 3 and 4 the whole left lung (Exp 1) or 1/5 of the left lung (Exp 3 and 4) were trimmed for histological examination and routinely paraffin wax embedded. Consecutive sections (3 μm) were prepared and routinely stained with hematoxylin-eosin (HE) or subjected to immunohistochemistry (IHC) for the detection of SARS-CoV-2 nucleoprotein (NP) [28] and Ly6G (neutrophil marker); for Exp 3, a further section of the infected lungs was stained for histone H3 (NET marker) [29]. All stains followed previously published protocols [30].

## Morphometric analyses

For quantification of SARS-CoV-2 antigen expression and the extent of neutrophil influx into the lungs, a morphometric analysis was undertaken on the slides stained for SARS-CoV-2 NP and Ly6G, respectively. The stained slides were scanned using NanoZoomer 2.0-HT (Hamamatsu, Hamamatsu City, Japan), and several sections of the lung of each animal were quantitatively analysed using the Visiopharm 2022.01.3.12053 software (Visiopharm, Hoersholm, Denmark). The average total tissue area used for quantification was $19.5 \pm 6$ $mm^2$. The morphometric analysis served to quantify the area, in all lung sections of an animal, that showed immunostaining for viral NP and Ly6G, respectively. In Visiopharm, for each section, the lung was manually outlined and annotated as a Region Of Interest (ROI), manually excluding artifactually altered areas. The manual tissue selection was further refined with an Analysis Protocol Package (APP) based on a Decision Forest classifier, with the pixels from the ROI being ultimately classified as either "Tissue" or "Background". This new "Tissue" ROI, regrouping the different lung samples analysed for each animal, was further quantified by executing two APPs successively. The first APP was based on a Threshold classifier and served to detect and outline areas with immunostaining. The second APP then measured both the surface of the immunostained area ($μm^2$) and the surface of the "Tissue" ROI ($μm^2$). The percentage of immunostained area (%), expressed as the ratio between the immunostained area and the total area, was obtained for each animal in Excel (Microsoft Office 2019; Microsoft, Redmond, Washington, United States), according to the following formula: ([positive area (μm2)]/ [total area (μm2)]) x 100.

## Statistical analyses

Statistical analysis was performed using GraphPad Prism 8.3 software (GraphPad Software, San Diego, CA, USA) and R software v3.6.3 (R core team). Statistically significant correlations between parameters were assessed by calculating Spearman's correlation coefficients, and differences between groups were assessed with Mann-Whitney, Kruskall-Wallis or ordinary one-way or 2-way ANOVA tests, depending on sample distribution and the number of groups analyzed. To elaborate, nonparametric tests like Mann-Whitney and Kruskall-Wallis were employed when the data violated assumptions of normality, while ANOVA tests were applied when the data met parametric assumptions.

## Results

### Unsupervised RNA-seq analysis reveals an antiviral gene expression signature of circulating neutrophils in COVID-19 that is strongly influenced by maturity

Different neutrophil subsets have gained a lot of attention as modulators of COVID-19 pathogenesis. We recently found increased frequencies of one neutrophil subset, referred to as low-density granulocytes (LDGs, isolated from the PBMC fraction), during COVID-19 [7]. In the current study we sought to understand in more detail the transcriptomic profile of different neutrophil subsets, including LDGs and their "normal" density counterpart, the circulating polymorphonuclear cells (PMNs) [31], which typically consists of mainly mature neutrophils. Neutrophils isolated from different cohorts comprised three PMN groups (severe COVID-19, mild COVID-19, and healthy controls), and one LDG group. Initial deconvolution of the RNA sequencing (RNA-seq) data allowed us to gain a comprehensive understanding of the cellular composition within PMN and LDG fractions and verified that most cells present in the samples were neutrophils (S1A Fig). This analysis also demonstrated that cells in the LDG fraction were predominantly immature neutrophils, meanwhile PMNs were composed of mainly mature neutrophils.

The samples with predominant neutrophil cell populations were selected for subsequent gene expression analysis (neutrophils $\geq$ 65%). The high variance in gene expression between PMNs and LDGs was confirmed by principal component analysis (PCA) (Fig 1A), which revealed that the gene expression patterns of COVID-19 LDGs differed from those of all PMNs regardless of the patients' disease state. Functional enrichment analyses through gene overrepresentation (ORA) and gene-set enrichment analyses (GSEA) (Fig 1B) compared PMNs with LDGs from severe COVID-19 patients. The most statistically significant result was an overrepresentation of the NOD-like receptor signaling pathway in PMNs in contrast with LDGs, highlighting that the different neutrophil fractions have a distinct inflammatory profile. This was supported by GSEA, where the most obvious increases in fold changes were the enrichment of the interferon signaling pathways. Another relevant difference was the cell cycle and DNA replication pathways, identified by both ORA and GSEA, which supported our previous findings suggesting LDGs to be predominantly immature cells [7]. Furthermore, a heatmap of selected type I IFN (IFN-I) related genes confirmed a robust IFN-I gene signature in severe COVID-19 PMNs, while LDGs from severe COVID-19 distinctively lacked this signature (Fig 1C). Unsupervised clustering analysis, namely Iterative Clustering and Guide Gene Selection (ICGS) using the AltAnalyze software, supported these findings by identifying the top 118 differentially expressed (DE) genes, including several IFN-related genes (S1B Fig). Similarly to the selected samples included in Fig 1, this analysis classified the samples into two major clusters: a first one containing all isolated LDG samples, and a second one comprising all isolated PMN samples. The former cluster consisted of neutrophil antimicrobial and granule marker genes (e.g. *DEFA3*, *DEFA4*, *SERPINB10*, *CTSG*), while in the latter cluster the most significantly upregulated genes in the PMNs from severe COVID-19 subgroup were mainly interferon inducible (e.g. *IFI44L*, *IFI6*, *GBP3*, *IRF7*). These differences were supported by a detailed gene analysis (S2A Fig).

### Increased expression of inflammasome related genes in severe COVID-19 PMNs

In addition to the strong IFN-I signature, PMNs of severe COVID-19 upregulated several genes involved in inflammatory processes, such as the formation of inflammasomes. We

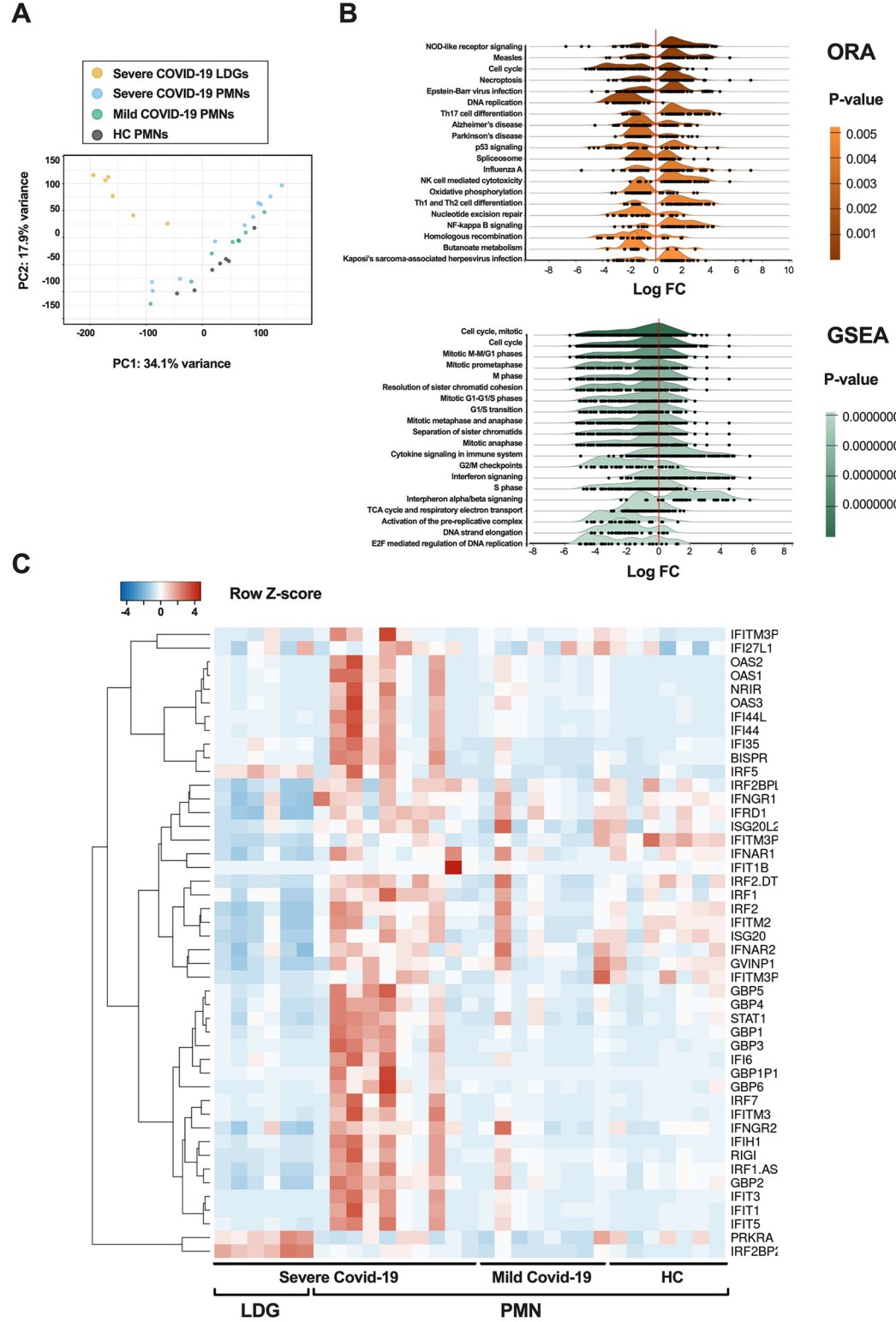

**Fig 1. Increased IFN-I related gene expression in mature COVID-19 neutrophils.** The analysis was reduced to include only the samples with the highest purity (cell fraction over 0.65 of neutrophils), as identified by CIBERSORTx. (**A**) Principal component analysis (PCA) of the RNA-seq samples (n = 7 PMNs from HC, n = 10 PMNs from severe COVID-19, n = 8 PMNs from mild COVID-19, and n = 6 LDGs from severe COVID-19). (**B**) Ridgeline diagrams depicting the top 20 enriched signal pathways from the genes differentially expressed by PMNs versus LDGs during severe COVID-19: overrepresentation analysis

(ORA) using KEGG database and gene-set enrichment analysis (GSEA) according to Reactome database. Both enrichment analyses were made using ExpressAnalyst and are sorted by P-value, obtained from Welch's t-test. (**C**) Heatmap of differentially expressed IFN-related genes in COVID-19 PMNs and LDGs as compared to HC PMNs. RNA sequencing was performed on purified PMNs from healthy controls, mild COVID-19 and severe COVID-19, as well as LDGs from severe COVID-19. The heatmap was clustered by complete linkage and ordered by Spearman's rank. *FC = fold change*.

further analyzed the differential expression of selected inflammasome related genes across all samples using RNA-seq (Fig 2A). PMNs of severe COVID-19 displayed higher levels of inflammasome genes such as NLRP3 and caspases 1, 4 and 5. LDGs did not display similar upregulation of inflammasome genes, with the notable exception IL-18 and NLRC4, which were not upregulated by PMNs. These findings prompted us to look more closely into PMN fractions between different disease states. Pathway analyses identified the inflammasome related NOD-like and RIG-like receptor signaling pathways among the most significantly overrepresented pathways, differentially expressed in severe COVID-19 PMNs versus HC PMNs (Figs 2B and S2B and S2C) or mild COVID-19 PMNs (Figs 2C and S2D and S2E). However, mild COVID-19 PMNs did not significantly differ from HC PMNs in their inflammatory profile (S2F Fig). The increased expression of selected IFN-I (*OAS1*, *OAS2*, and *IFIT1*) and inflammasome related genes (*CASP1*, *CASP5*, *NLRC5* and *NAIP*) between COVID-19 and HC PMNs was confirmed by RT-qPCR. However, some inflammasome related genes (*IL-1β*, *NLRP3* and *NLRC4*) were seemingly downregulated, although not statistically significant (S3 Fig).

Single cell sequencing data from the COVID-19 immune atlas, which integrates data from 5 independent studies analyzing COVID-19 PBMCs, confirmed our transcriptomic results (Fig 2D), from which a detailed gene by gene analysis of the most relevant inflammasome related genes is shown (Figs 2E and S4). Briefly, PYCARD gene coding for the ASC protein was expressed similarly in mature and immature neutrophils (S4 Fig), suggesting that both cell types may have ASC-dependent inflammasome forming capacity. However, most of the inflammasome gene expressions differed significantly and in the same manner as in our transcriptomic analysis.

## Activation of neutrophil inflammasome related pathways during respiratory distress is not specific to COVID-19

We also reanalyzed the RNA-seq data generated by LaSalle *et al*. [8], focusing on neutrophil transcriptomics in patients with COVID-19 versus non-COVID-19 patients, and healthy controls. The non-COVID-19 patients exhibited acute respiratory distress and clinical suspicion for COVID-19. However, they tested negative for SARS-CoV-2 by PCR, unlike those classified as COVID-19 patients. Our analysis included IFN-α response, IL-1β production, TLR signaling, NLRP3 inflammasome, and pyroptosis pathways, using the Gene Ontology (GO) database; the NLR signaling pathway using the Kyoto Encyclopedia of Genes and Genomes (KEGG) database; and inflammasome pathway using the REACTOME database (Fig 3). These pathways were significantly enriched in COVID-19 patients, supporting our findings. Importantly, the genes from the above-mentioned pathways were also induced in non-COVID-19 patients, suggesting that these pathways represent a general neutrophil response to inflammatory stimuli rather than a COVID-19 specific response.

## Inflammasomes are activated in severe COVID-19 PMNs, but not directly by SARS-CoV-2

Given the strong upregulation of many inflammasome related genes during severe COVID-19, we assessed whether PMNs exhibit active inflammasome formation *in vivo*. To evaluate

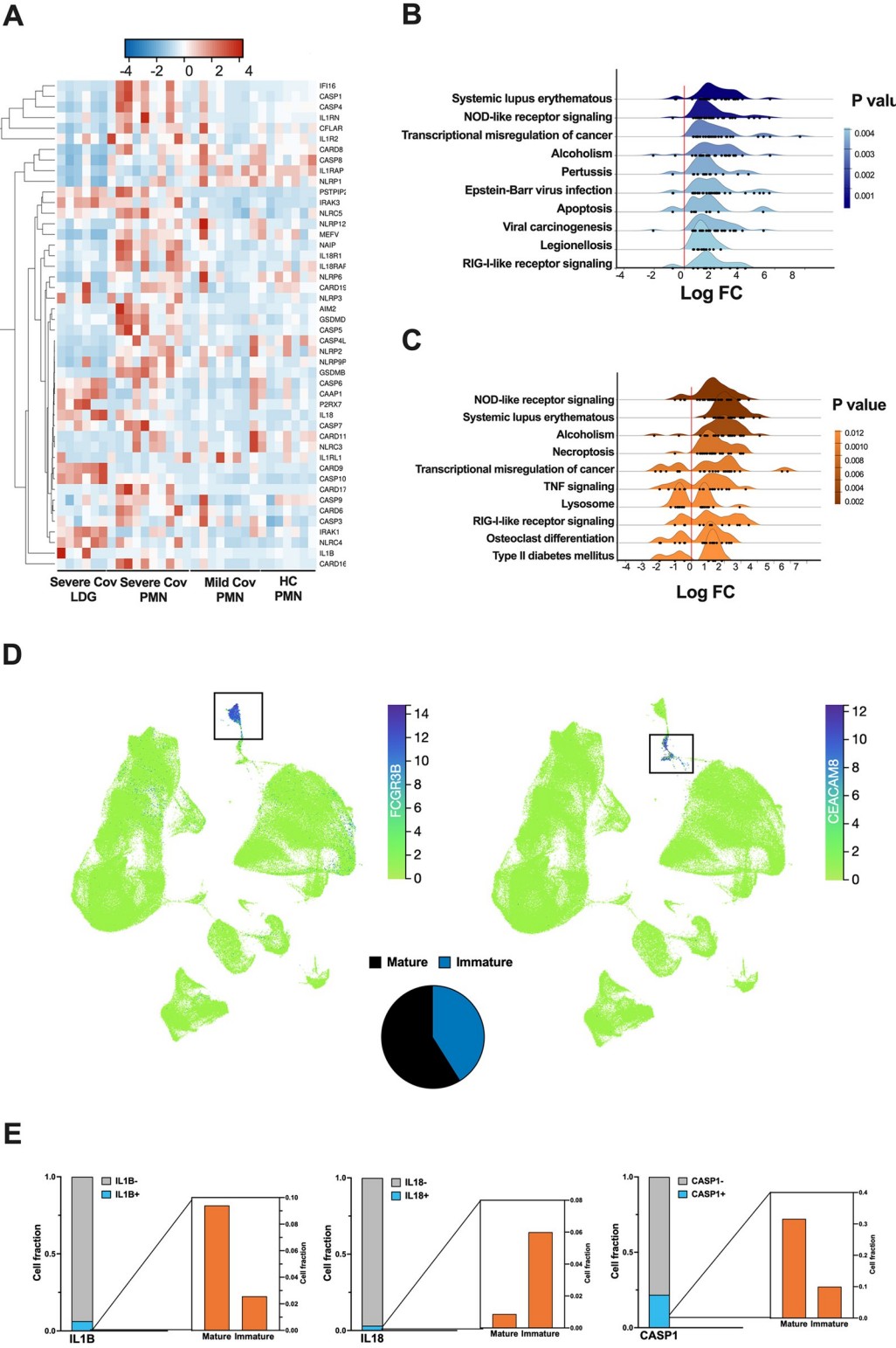

**Fig 2. Inflammasome related gene expression.** (**A**) Heatmap depicting selected differentially expressed inflammasome related genes from RNA sequencing performed in PMNs from HC, mild and severe COVID-19, as well as severe COVID-19 LDGs. Only the samples with the highest purity, determined by a cell fraction over 0.65 of neutrophils (identified by CIBERSORTx) are included. The heatmap was clustered by complete linkage and ordered by Spearman's rank. (**B-C**) Ridgeline diagrams of overrepresentation analyses (ORA) according to KEGG database, depicting the top 10 enriched

signaling pathways in PMNs during severe COVID-19 compared to (**B**) healthy controls and (**C**) mild COVID-19. (**D**) UMAP analysis of the COVID-19 Immune Atlas, which integrates 5 public COVID-19 PBMC single-cell transcriptomics datasets, created using CELLxGENE. (Top) UMAP showing the clustering of CD16+ cells (mature, FCGR3B expressing cells) and CD66b+ cells (immature, CEACAM8 expressing cells). Each dot represents a single cell colored according to the expression level of a selected gene. The color scale ranges from green (low expression) to purple (high expression). (Bottom) Pie chart summarizing the percentage of mature (black) and immature (blue) cells in the data. (**E**) The fraction of mature and immature neutrophils cells expressing inflammasome related genes identified in (**D**) are shown in bar graphs. For each gene, the proportion of expressing cells is shown in light blue, while the proportion of negative or not-expressing cells is shown in gray. Zoomed-in bar graph depicts the proportion of mature and immature cells expressing each gene.

spontaneous inflammasome mediated cytokine secretion, fresh PMNs isolated from severe COVID-19 patients and HC were cultured *ex vivo* for 24 hours. We measured the levels of IL-1β and IL-18 in the supernatant and found that IL-1β secretion was significantly increased in the supernatant of severe COVID-19 PMNs compared to HC PMNs (Fig 4A), whereas the IL-18 levels did not differ significantly (Fig 4B). Additionally, since SARS-CoV-2 viral particles were previously implicated to induce inflammasome formation in macrophages [17], the IL-1β and IL-18 levels after HC PMNs exposure to SARS-CoV-2 were also assessed but no significant effects in the secretion of these cytokines were observed (Fig 4C and 4D).

The spontaneous secretion of IL-1β by COVID-19 PMNs suggests that these cells are actively producing and releasing IL-1β through inflammasome formation which is dependent on caspase1 activity [32]. We assessed caspase1 activity in response to the second signal required for inflammasome activation, induced by nigericin, and observed increased caspase1 activity in severe COVID-19 PMNs compared to HC PMNs (Fig 4E). These findings suggest that severe COVID-19 PMNs have an increased capacity for inflammasome activation, potentially due to an existing priming signal during acute disease *in vivo*. However, no significant difference in caspase1 activity between non-exposed and virus-exposed PMNs were observed (Fig 4F), indicating that caspase1 activation in COVID-19 PMNs is not directly triggered by the virus.

## Type I IFNs prime PMNs for inflammasome activation

Since PMNs from COVID-19 patients concomitantly display a strong IFN-I signature (Fig 1B and 1C) and an increased propensity for inflammasome activation, we hypothesized that IFN-I could act as the priming signal for PMN inflammasomes during COVID-19. Isolated HC PMNs were stimulated *ex vivo* with exogenous IFN-I and the well-described inflammasome priming (1st signal) and activator (2nd signal) agents LPS and nigericin, respectively [33,34]. After stimulation, both priming signals induced pro-IL-1β (31 kDa) in the cell lysates, followed by the release of active IL-1β (17 kDa) into the supernatant in response to nigericin (Fig 5A), confirming the ability of IFN-I to prime PMNs for inflammasome activation.

To assess inflammasome formation in circulating neutrophils during COVID-19, PMNs from HC and COVID-19 patients underwent similar stimulation assays as above, followed by IL-1β measurement from supernatants by ELISA. In addition, to further assess the role of SARS-CoV-2 virus particles in neutrophil inflammasome activation, HC PMNs were cultured in the presence of purified viruses (10 infectious units/PMN). HC PMNs responded to both LPS and IFN-I by increasing their IL-1β secretion, which was exponentiated after exposure to nigericin (Fig 5A and 5B), confirming the ability of IFN-Is to prime for inflammasome assembly in PMNs, albeit less efficiently than LPS. Furthermore, as expected, the ability of IFN-I to prime for nigericin-mediated inflammasome activation was dependent on the IFN-I receptor IFNAR1 for both IFN-α and IFN-β (S5A Fig).

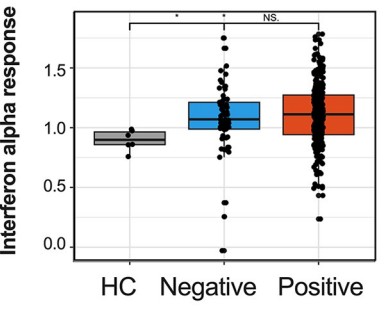

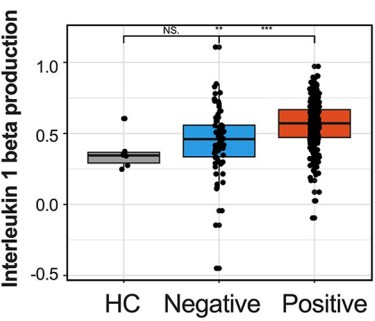

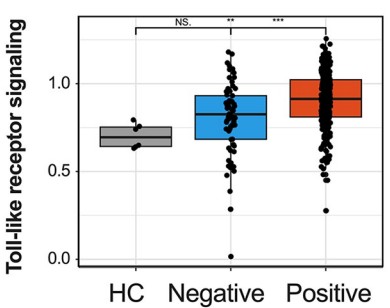

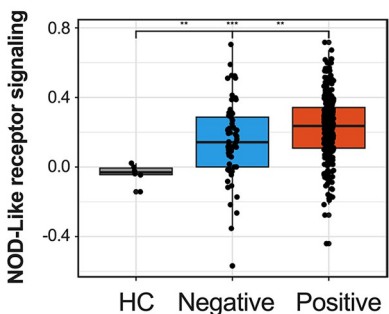

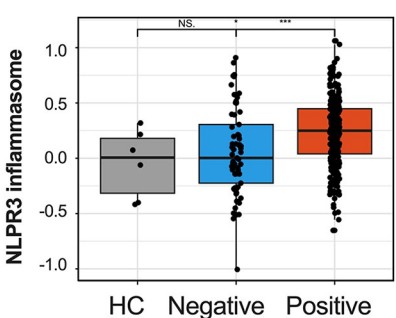

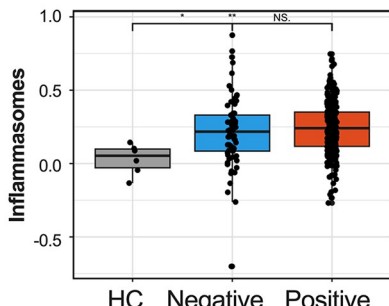

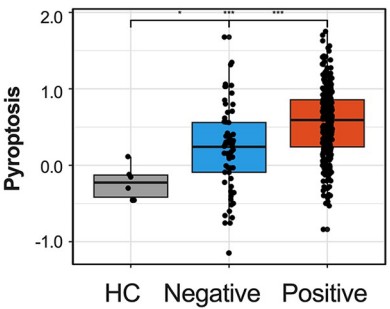

**Fig 3. Comparative neutrophil transcriptomics of COVID-19 and non-COVID-19 patients.** Bar graphs represent the activation levels of selected pathways and processes as identified by neutrophil transcriptomics. The analysis includes interpheron alpha (IFN-α) responses, interleukin (IL)-1β production, Toll-like receptor (TLR) signaling, NLRP3 inflammasome, and pyroptosis, as determined through the Gene Ontology (GO) database. The NOD-like receptor signaling pathway was investigated using the Kyoto Encyclopedia of Genes and Genomes (KEGG) database,

and the inflammasome pathway was explored via the REACTOME database (more information in S1 Table). The graphs compare the activation levels of these pathways in healthy controls (HC), non-COVID patients with similar symptoms (COVID-19 negative), and COVID-19 positive individuals. Statistical significance is denoted as follows: *p < 0.05, **p < 0.01, ***p < 0.001, ****p < 0.0001. P values were calculated with Kruskall-Wallis test.

Interestingly, COVID-19 PMNs produced less IL-1β than HC PMNs upon exogenous inflammasome activation primed by either LPS or IFN-I, while SARS-CoV-2 particles did not have any effect on PMN inflammasome activation (Fig 5B). As with 24 h cultures (Fig 4B), we did not detect any significant changes in IL-18 secretion in either HC or COVID-19 PMNs (S5B Fig). However, the release of myeloperoxidase (MPO), used as a marker of degranulation and/or NETosis, in response to nigericin was similar between COVID-19 PMNs and HC PMNs (Fig 5C), and therefore the observed diminished IL-1β release by COVID-19 PMNs is not due to general cellular inertia but may be specific to the *ex vivo* induced inflammasome pathway. Furthermore, additional stimulation assays in the presence of the NLRP3 inhibitor MCC950 (Fig 5D and 5E) and caspase1 inhibitor YVAD (S5C Fig) confirmed that induced IL-1β secretion is dependent on canonical NLRP3 inflammasome activation. Unlike IL-1β (S5D Fig), increased IL-18 secretion was not detectable even after 24 h stimulation (S5E Fig). Furthermore, the observed residual IL-18 was not affected by inflammasome inhibitors, suggesting its secretion to be unrelated to inflammasome activity in PMNs.

We further assessed the specificity of inflammasome activation by measuring LDH and IL-8 levels in the supernatants from the same cells and under the same experimental conditions as shown in Fig 5D and 5E. The measurements of the former were done to assess inflammasome mediated cell death by pyroptosis in response to nigericin, while the latter was assessed to demonstrate the responsiveness of PMNs to an inflammasome unrelated inflammatory cascade. As with IL-1β secretion, COVID-19 PMNs were less responsive than HC PMNs to nigericin- and LPS-mediated LDH (Fig 5F) and IL-8 (Fig 5G) release, respectively. This suggests that COVID-19 PMNs are generally poorly responsive to inflammatory stimuli.

To examine this reduced responsiveness to external inflammatory priming, we evaluated the inflammasome related gene expression following *ex vivo* stimulation with IFN-I or LPS (Figs 5H–5K and S5F–S5I). OAS1 gene, an interferon stimulated gene (ISG), showed significant upregulation by IFN-I in COVID-19 PMNs as compared to HC PMNs (Fig 5H), while the inflammasome related genes IL-1β (Fig 5I), CASP1 (Fig 5J) and NLRC5 (Fig 5K) were more efficiently induced in HC PMNs than COVID-19 PMNs. This suggests that the inflammasome defect in COVID-19 PMNs is at the transcriptional level when using IFN-I as the priming factor, while high OAS1 gene expression indicates transcriptional defect is restricted to individual genes.

Next, we investigated whether IFN-I could also activate caspase1 directly without the 2nd signal to boost inflammasome activation. In addition to treating HC PMNs with IFN-I and LPS as in previous experiments, we also included another group with a higher dose of IFN-I (ten-fold) and assessed caspase1 activity after 4 hours with FAM-FLICA, a fluorescent caspase1 reactive dye (Fig 5L) as well as IL-1β release after 24 hours by ELISA (Fig 5M). Results showed that similar to LPS, high-dose IFN-I induced significant caspase1 activity, which was not observed with the normal IFN-I concentration. Despite this, the normal IFN-I concentration still resulted in increased IL-1β levels in the supernatant. Taken together, these findings suggest that the 2nd signal is not essential for inflammasome activation by IFN-I and that IL-1β release is a more sensitive method of detecting inflammasome activity as compared to caspase1 activity in our assay setup. Thus, these results can also explain our previous observation of

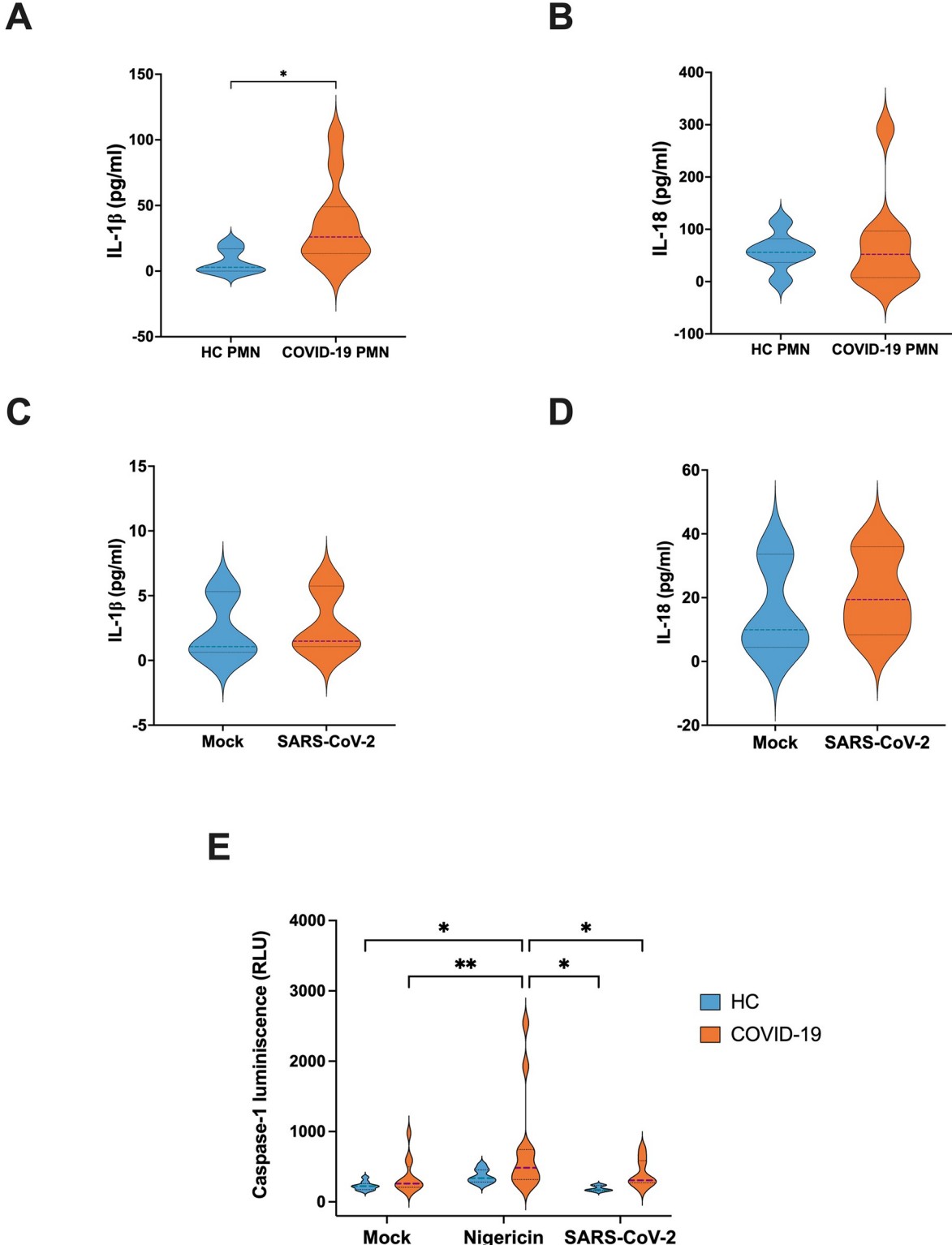

**Fig 4. Inflammasome activation in PMNs during severe COVID-19.** PMNs were freshly isolated from blood and cultured at 2 million cells/ml (**A**) IL-1β and (**B**) IL-18 levels in 24 h cell culture supernatants from COVID-19 (n = 11 for IL-1β and 9 for IL-18) and HC PMNs (n = 6 for both). (**C**) IL-1β and (**D**) IL-18 levels in 24 h cell culture supernatant from PMNs exposed or non-exposed to purified SARS-CoV-2 viral particles (10 virus particles / PMN) (n = 3). (**E**) Caspase1 activity in PMNs following a 2 h stimulation with nigericin or purified SARS-CoV-2 viral particles (10 virus particles / PMN). For HC PMNs, n = 9 for mock and nigericin and n = 6 for SARS-CoV-2

exposure. For COVID-19 PMNs, n = 12 for mock and nigericin and n = 9 for SARS-CoV-2 exposure. *p < 0.05 and **p < 0.01. Data presented as mean ± SD. Tukey's multiple comparisons test for mixed-effect analysis was applied for (**E**), meanwhile P values for (**A-D**) were calculated with the Mann-Whitney U-test.

increased spontaneous release of IL-1β by COVID-19 PMNs even though significant caspase1 activity is only detected after nigericin-mediated boosting *in vitro* (Fig 4A and 4E).

## Association between *ex vivo* inflammasome activation and disease severity

Our analysis of the association between *ex vivo* inflammasome activation (caspase1 activity and IL-1β release) and clinical markers of disease severity, including neutrophil responses, revealed intriguing links. Calprotectin is a marker of neutrophil activation or death [35] but also potentially activates the inflammasome [36]. A significant positive correlation between calprotectin plasma levels and PMN caspase1 activity (Fig 6A and 6B) underscores this latter possibility and highlights the interplay between inflammation and inflammasome activation in PMNs of COVID-19 patients. Furthermore, the negative association of PMN IL-1β levels (after *ex vivo* stimulation with IFN and nigericin) with disease severity (WHO ordinal scale, Fig 6A) and patient neutrophil counts (Fig 6A and 6C) supports the exhaustion hypothesis, wherein PMNs from severe COVID-19 patients may be less responsive to stimuli due to prior *in vivo* activation. While these findings provide intriguing insights into the complex interplay between calprotectin release, caspase1 activity, and inflammasome activation in COVID-19, additional research is required to further elucidate these connections.

## LDGs differ from PMNs in their ability to release IL-18 upon inflammasome activation

Transcriptomic analysis presented above revealed a distinct lack of IFN-I responsive and inflammasome related gene expression in LDGs as compared to PMNs of severe COVID-19 patients. This suggested that inflammasomes are not similarly regulated in LDGs as compared to PMNs during COVID-19. To assess the inflammasome forming capacity of LDGs, we conducted *ex vivo* stimulation assays using LDGs isolated from COVID-19 patients, similar to the approach used for PMNs described earlier. Like PMNs, IL-1β secretion by LDGs was elevated in the presence of a priming signal (IFN-I or LPS), which exponentially increased when the inflammasome activation signaling molecule nigericin was added (Fig 7A). Contrary to PMNs and in line with the transcriptomics data, an increased IL-18 secretion was detected (Fig 7B). Additionally, the secretion of both ILs by LDGs was inhibited in the presence of inflammasome specific inhibitors MCC950 and YVAD (Fig 7A and 7B).

These findings suggested that IFN-I can prime for inflammasome activation also in LDGs. Furthermore, the ability to release IL-18 upon neutrophil inflammasome activation varies based on cellular maturation state. To explore this further, we conducted *in vitro* stimulation studies using differentiated HL-60 cells, an immature neutrophil-like model [37]. Similar to LDGs from COVID-19 patients, HL-60 displayed comparable IL-18 secretion pattern upon LPS or IFN-I stimulation and nigericin-induced activation. Notably, their IL-1β release was only detected with LPS priming (Fig 7C and 7D). Furthermore, transcriptomic analysis revealed an upregulation of inflammasome related genes upon differentiation (Fig 7E). Overall, these findings suggest that neutrophils may lose the ability to secrete IL-18 in response to inflammasome activity during maturation, and increased release of neutrophil-derived IL-18 occurs primarily in disease states associated with extensive granulopoiesis and increased immature granulocyte counts in the blood, like COVID-19 [38].

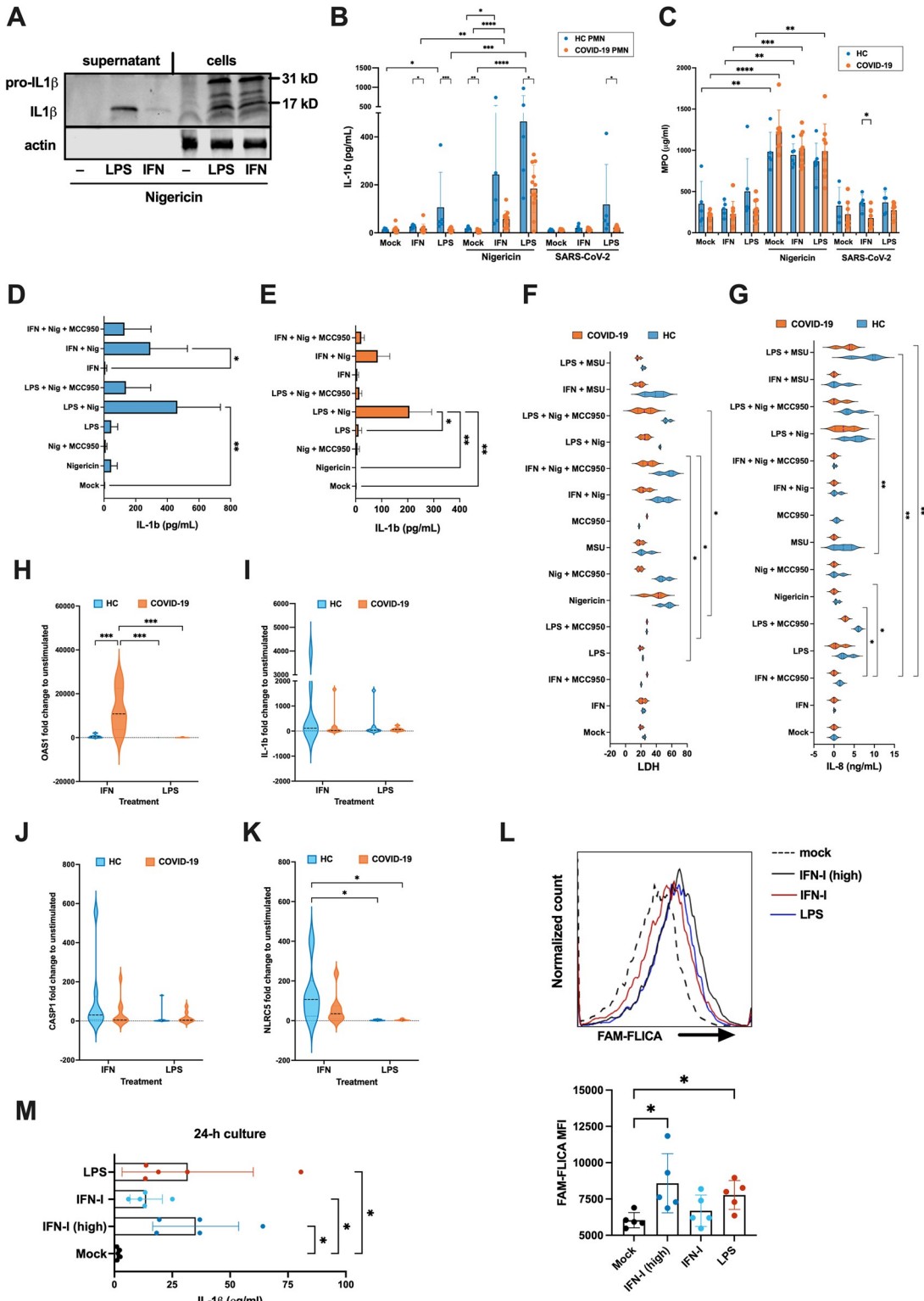

**Fig 5. IFN-I primes inflammasome activation while COVID-19 PMNs show defective inflammasome responses *ex vivo*.** Isolated HC or COVID-19 PMNs were non-stimulated or stimulated 4h with IFN-I (combination of 2.7*10^4 IU/ml IFN-α and IFN-β) or 20 ng/ml LPS (1st signal), followed by 4h with 2.5 μM nigericin or purified SARS-CoV-2 (10,1 virus/PMNs) (2nd signal). Then, (**A**) western blot of pro-IL-1β (31 kD) and active IL-1β (17 kD) was performed from HC PMNs supernatant and cell lysates, (**B**) IL-1β (n = 5 HC PMN and 9 COVID-19 PMN) and (**C**) MPO (n = 5 HC PMN and 9 COVID-19 PMN) were

measured from supernatants by ELISA. (**D-E**) Effect of inflammasome inhibitor MCC950 (2 μg/ml, added simultaneously with nigericin) on IL-1β secretion in (**D**) HC and (**E**) severe COVID-19 PMN supernatant (n = 3). (**F**) LDH and (**G**) IL-8 in HC and severe COVID-19 PMN supernatants (n = 3). (**H-K**) RT-qPCR of selected mRNAs in IFN-I or LPS-primed HC and COVID-19 PMNs (n = 6–8 HC PMN and 7–10 COVID-19 PMN). (**L-M**) HC PMNs were stimulated with high dose IFN-I ($2.7*10^5$ IU/ml), normal dose IFN-I ($2.7*10^4$ IU/ml) and 20 ng/ml LPS. After 4 hr stimulation caspase1 activity was measured using median fluorescence intensity (MFI) of FAM-FLICA by flow cytometry (**L**, n = 5, representative histogram of one donor shown) and after 24 hr stimulation IL-1β release was measured by ELISA (**M**, n = 5). P values calculated with Kruskall-Wallis test for the comparison between treatments by group (HC or COVID-19 PMNs), and Mann-Whitney test for the comparison between HC and COVID-19 PMNs by individual treatment for (**B-G**), and Two-way ANOVA Tukey's multiple comparisons test for (**B, H-K**). The treatments in L-M were compared to mock by one-way ANOVA for repeated measures, corrected for multiple comparisons with the two-stage step-up method of Benjamini, Krieger and Yekutieli. *$p < 0.05$, **$p < 0.01$, ***$p < 0.001$, **** $p < 0.0001$. Data presented as mean ± SD.

## Neutrophils are recruited to the lungs in SARS-CoV-2 infected mice

Hamsters and human ACE2 expressing mice infected with SARS-CoV-2 develop pulmonary inflammation including neutrophil recruitment [39–41]. To further assess the role of neutrophils in COVID-19, we utilized a recently developed SARS-CoV-2 mouse model [18]. This model employs the MaVie strain, serially passaged in mouse lungs and causing pneumonia like human COVID-19 in wild-type BALB-C mice [18]. Infected mice started losing weight by day 2 post-infection, with some mice reaching the clinical endpoint of 20% weight loss by day 4 (S6A Fig, includes animals from 4 independent infection experiments, details of animal usage in S3 Table). The first experiments were performed to study infection kinetics, histopathology and Ly-6G+ neutrophil accumulation in lungs.

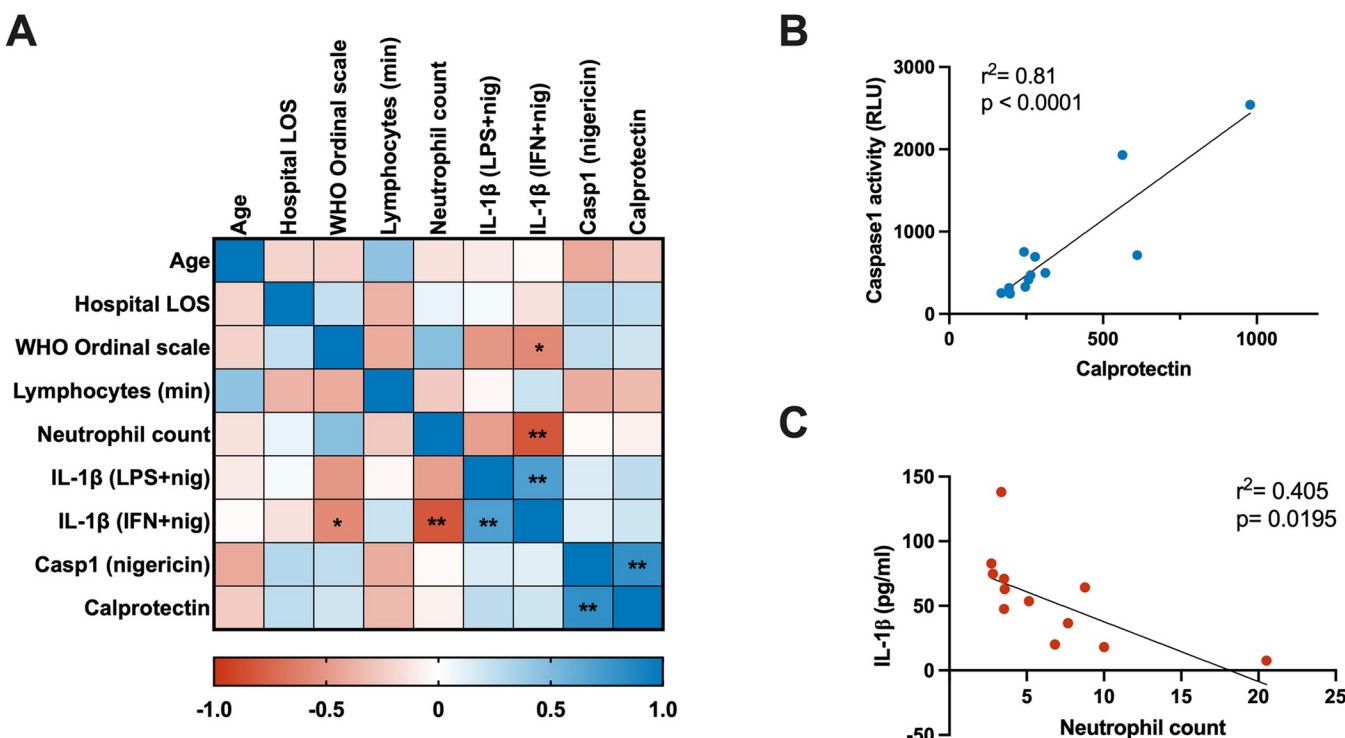

**Fig 6. Correlation analysis between clinical parameters and *ex vivo* PMN inflammasome activation.** (**A**) Spearman's correlation matrix depicting the relationships among clinical parameters and results of *ex vivo* experimentation. For the WHO ordinal scale, the baseline parameters were used. (**B-C**) Linear regression analysis demonstrating the associations between: (**B**) Positive association between PMN Caspase1 activity, measured after *ex vivo* nigericin stimulation, and the levels of Calprotectin in the matched patient's peripheral blood; (**C**) Negative association between ex vivo stimulated PMN IL-1β levels (IFN+Nig) and the blood neutrophil count in matched patients at the time of sampling (n = 12). *LOS = length of stay. WHO = World Health Organization. Min = minimum. Casp1 = caspase1. LPS or IFN + nig = lipopolysaccharide or type I interferon + nigericin ex vivo stimulation.*

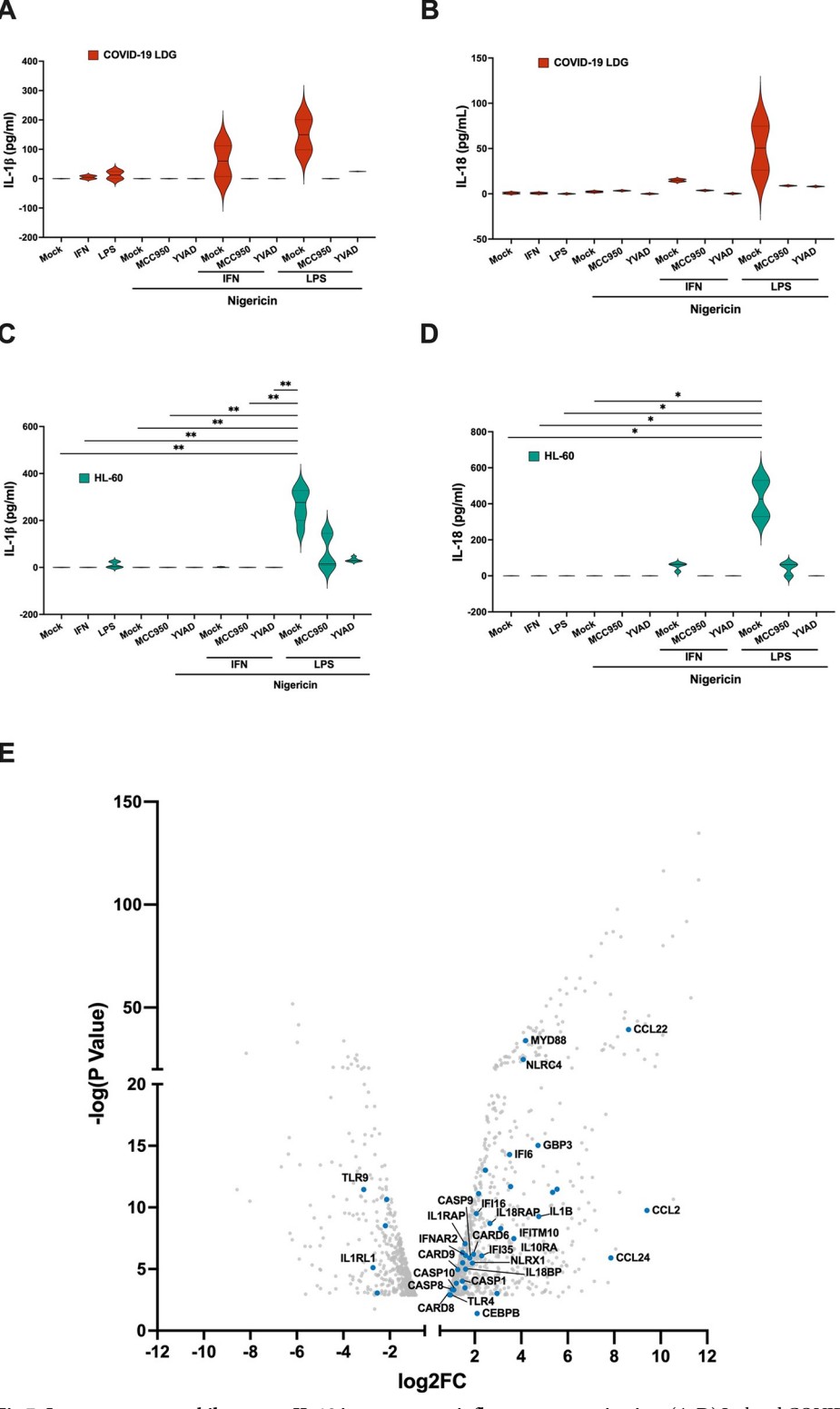

**Fig 7. Immature neutrophils express IL-18 in response to inflammasome activation.** (**A-D**) Isolated COVID-19 LDGs or HL-60 cells (differentiated for 5 days with 1% DMSO) were non-stimulated or stimulated 4h with IFN-I or LPS (1st signal), followed by 4h with nigericin (2nd signal) in the presence or absence of inflammasome inhibitors MCC950 or YVAD as previously. Secretion of (**A, C**) IL-1β and (**B, D**) IL-18 were measured from the supernatants by ELISA (n = 2 for LDGs and 3–5 for HL-60). *p < 0.05 and **p < 0.01. P values calculated with Kruskal-Wallis test.

Data presented as mean ± SD. (**E**) Volcano plot of differentiated vs undifferentiated HL-60 cells gene expression from GSE93996, with inflammasome related genes marked in blue. Only significant DE genes are shown (adjusted p value < 0.05).

Viral loads, as assessed by RT-qPCR and titration of infectious virus, were significantly higher at 2 dpi than at 4 dpi (Fig 8A and 8B), and viral antigen expression, widespread at 2 dpi in bronchioles and alveoli, matched this pattern (Figs 8C and S6D). The extensive viral replication at 2 dpi was associated with degeneration of infected epithelial cells, most prominent in the respiratory epithelium, accompanied by neutrophil (Ly6G+) infiltration (S6D Fig). An increase in neutrophil numbers in the lungs of infected mice was observed as represented by increased Ly6G+ neutrophil/lymphocyte ratio and total Ly6G+ cell counts as assessed by flow cytometry of lung single cell suspension (Fig 8D and 8E; the gating strategy for Ly6G+ neutrophils is shown in S6B Fig). A significant increase in the number of neutrophils in the lungs of infected mice compared to PBS-inoculated mice was also confirmed *in situ*, by morphometrical quantification of lungs sections stained for Ly6G (Figs 8F and S6D). Detailed information on the histological and immunohistochemical features is provided in S3 Table.

Furthermore, flow cytometry revealed diminished surface expression of the maturation and activation marker CD11b in Ly6G+ neutrophils of infected mice both at 2 and 4 dpi as compared to PBS controls (Fig 8G). However, morphologically, neutrophils from infected mice were similar to those in the PBS control mice in displaying equally multilobed nuclei (DNA staining of isolated Ly6G+ neutrophils shown in Fig 8H; a representative flow cytometry histogram showing the purity of Ly6G+ neutrophils after isolation is provided in S6C Fig). This suggests that the reduced expression of CD11b on the surface of neutrophils from infected mice was not the consequence of an accumulation of CD11b-negative immature neutrophils but rather an activation-related phenomenon, such as shedding or internalization.

## Neutrophils from SARS-CoV-2 infected mice display IFN-I dependent caspase1 activation

Next, we investigated whether neutrophils of SARS-CoV-2 infected mice show increased caspase1 activation. We utilized the fluorescent FAM-FLICA caspase1 reactive probe in conjunction with Ly6G+ neutrophil staining to detect caspase1 activity in neutrophils by flow cytometry. Indeed, we observed significantly an increased median fluorescent intensity of FAM-FLICA in Ly6G+ neutrophils at 4 dpi, but not at 2 dpi, compared to PBS-inoculated controls (Fig 9A). This finding suggests increased neutrophil caspase1 activity at latter stages of the infection, concomitant with decreased viral loads.

The potential role of IFN-I in mediating caspase1 activity in neutrophils of infected mice was initially assessed by isolating neutrophils from the lungs of infected mice at 2 and 4 dpi, as well as from non-infected mice, for transcriptomic analysis by RNA-seq. PCA showed differences between neutrophils from infected and non-infected mice, with slight variation between the 2 and 4 dpi time points (Fig 9B). These differences were reflected in many DEGs, including several IFN-I responsive and inflammasome related genes, which showed strong upregulation at 2 dpi with slightly lower but still significantly elevated levels at 4 dpi, compared to non-infected mice (highlighted in the DEG heatmap; Fig 9C). The volcano plot (Fig 9D) provided a comprehensive view of the DEG pattern between neutrophils from SARS-CoV-2 infected and mock-infected mice. In addition to confirming the upregulation of IFN-I responsive and inflammasome related genes observed in the heatmap, the plot revealed a broader transcriptional response to viral infection with several additional DEG.

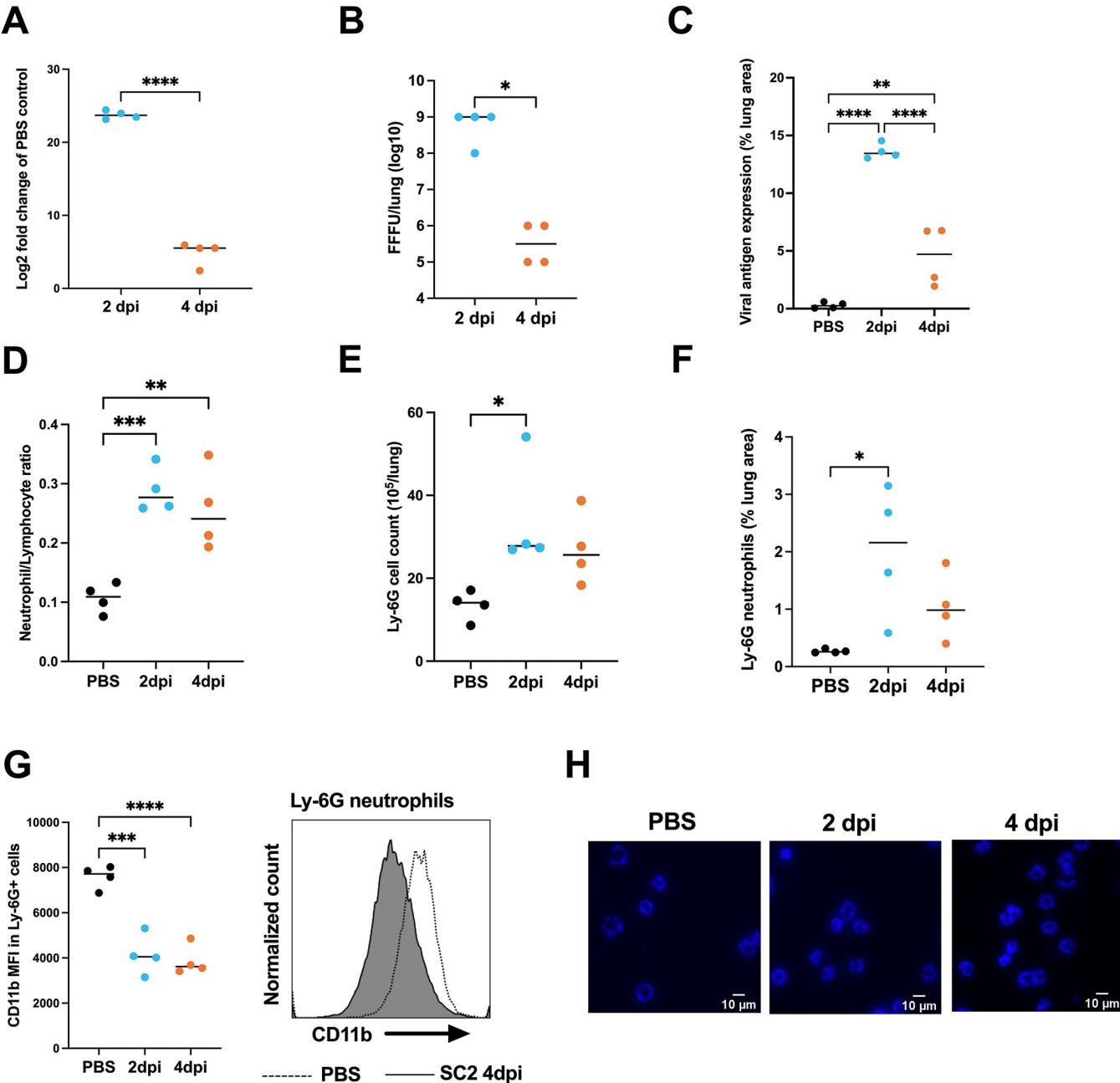

**Fig 8. Neutrophil accumulation in the lungs correlates with viral loads in SARS-CoV-2 infected mice.** Female BALB/c mice were intranasally inoculated with $5*10^5$ TCID50 SARS-CoV-2 MaVie strain or PBS as control and euthanized at 2 dpi or 4 dpi. (**A**) RNA was isolated from lungs and subjected to RT-qPCR targeting viral subE and GAPDH as housekeeping gene. The relative expression of subE was measured using the comparative Ct method as compared to mock-infected control (in which subE was undetectable but set to 40 Ct) **** $p < 0.0001$. P values calculated with Welch's t-test. (**B**) Infectious virus was calculated from supernatants of lung single cell suspensions of infected mice as fluorescence focus forming units (FFU) in Vero E6 cells. * $p < 0.05$. P values calculated with Mann-Whitney test. (**C**) Quantification based on morphometric analysis that determines the area of immunolabelling for SARS-CoV-2 nucleoprotein in relation to total tissue area. (**D**) Quantification of Ly6G neutrophil/lymphocyte ratio in lung single cell suspensions by flow cytometry. (**E**) Quantification of total Ly6G neutrophil counts in lung single cell suspensions by flow cytometry. (**F**) Quantification of Ly6G based on morphometric analysis that determines the area of immunolabelling for Ly6G in relation to total tissue area in mock-infected controls. (**G**) Quantification of median fluorescence intensity (MFI) of CD11b expression in ly6G neutrophils by flow cytometry. Representative histograms of CD11b expression in Ly6G+ neutrophils is shown. P values for C-G were calculated with ordinary one-way ANOVA using Tukey's multiple comparison correction. Black line represents the mean. * $p < 0.05$, ** $p < 0.01$, *** $p < 0.001$, **** $p < 0.0001$. (**H**) Fluorescent nuclear staining of representative magnetic-bead isolated Ly6G neutrophils by Hoechst33342. Panels A, C and F are representative of two independent experiments.

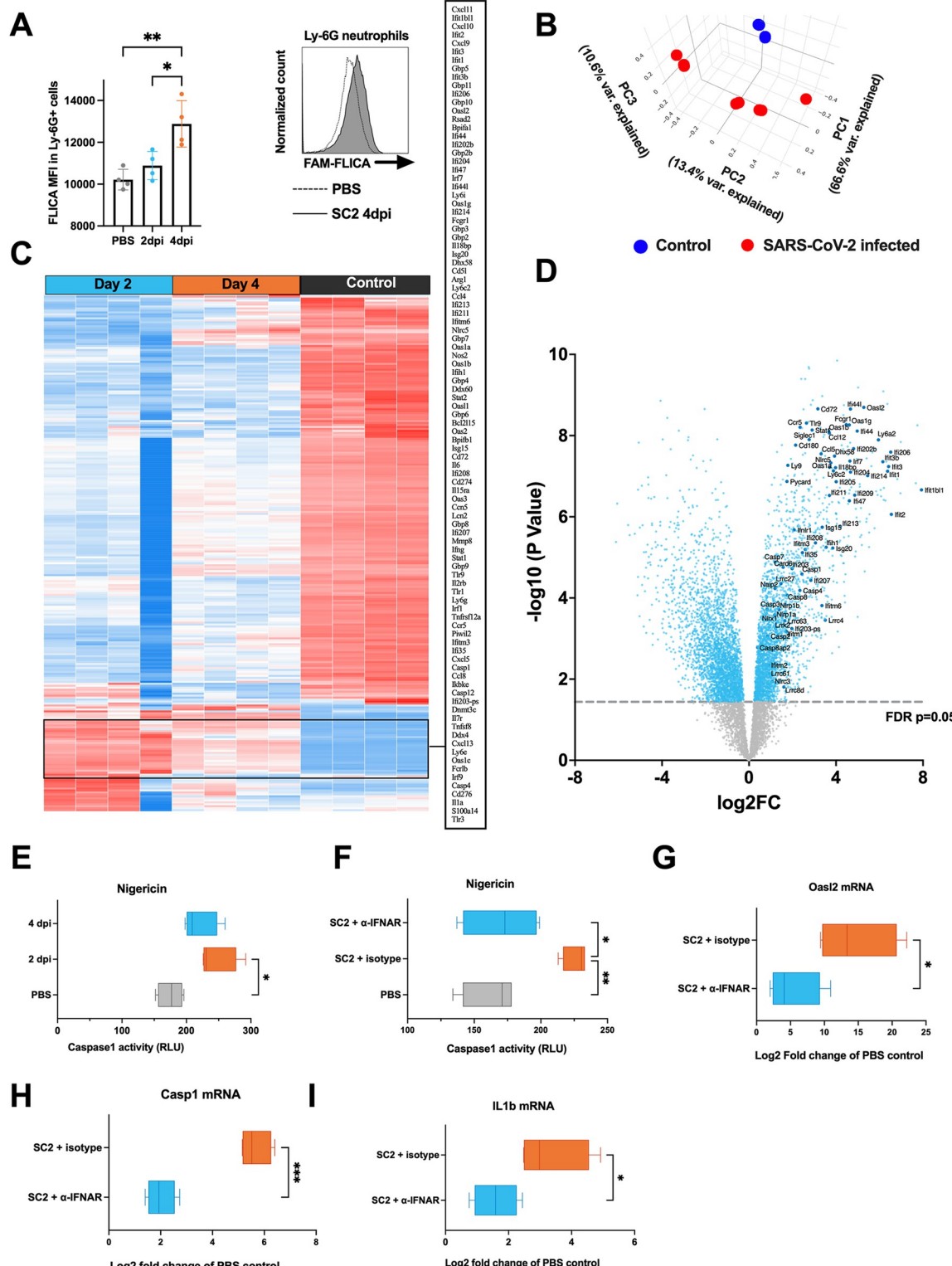

**Fig 9. Neutrophils of SARS-CoV-2 infected mice display increased caspase1 activation ability that is dependent on IFN-I.** Female BALB/c mice were intranasally inoculated with 5* $10^5$ TCID50 SARS-CoV-2 MaVie strain or PBS as control. (**A**) Lungs were harvested at 2 and 4 dpi and single cell suspensions stained with FAM-FLICA and Ly6G antibody followed by flow cytometric analysis. FAM-FLICA median fluorescence intensity (MFI) was recorded in Ly6G+ neutrophils (n = 4, representative histogram image shown). * p < 0.05, ** p < 0.01. P values calculated with one-way ANOVA using Tukey's multiple comparisons. (**B-E**) Ly-6G+ neutrophils

isolated from lung single cell suspensions based on positive selection with magnetic beads. (**B-D**) RNA was isolated and subjected to transcriptomic analysis by RNA-seq. (**B**) Principal component analysis (PCA) of the PBS-inoculated control and SARS-CoV-2 infected mice lung neutrophil RNA-seq samples. (**C**) Heatmap of the top differentially expressed genes (DEGs). (**D**) Volcano plots of DEGs between neutrophils isolated from SARS-CoV-2 infected mice versus uninfected PBS-inoculated mice. Blue points represent significant terms (adjusted p-value < 0.05), while smaller gray points represent non-significant terms. Relevant inflammasome and interferon related genes are shown with larger and darker blue points. (**E**) Caspase1 activity in isolated mice neutrophils following a 2 h stimulation with nigericin was assessed by a bioluminescence method (Caspase-Glo 1 Inflammasome Assay). (**F-I**) Mice were intraperitoneally inoculated with 250 µg anti-IFNAR or IgG1 isotype control directly after infection with SARS-CoV-2 and lung neutrophils isolated at 2 dpi (including also intranasally PBS-inoculated control mice without intraperitoneal injection). (**F**) Caspase1 activity was assessed following a 2 h stimulation with nigericin by bioluminescence method. (**G-I**) RNA was isolated from isolated neutrophils and fold change mRNA expressions of (**G**) Oasl2, (**H**) Caspase1 (Casp1) and (**I**) IL-1β (Il1b) was assessed by RT-qPCR in isotype control and anti-IFNAR treated infected mice as compared to mock-infected control mice. *p < 0.05, **p < 0.01 and ***p < 0.001. P values for A, E and F panels were calculated with ordinary one-way ANOVA using Tukey's multiple comparisons correction, while Welch's t-test was used for panels G-I. Data presented as mean ± SD.

Having established a robust IFN-I transcriptional signature in neutrophils from SARS-CoV-2 infected mice we wanted to assess whether they possess increased propensity for nigericin-induced caspase1 activation, similar to human PMNs after exogenous priming by IFN-I. Neutrophils from infected mice, harvested at 2 and 4 dpi, displayed increased caspase1 activity upon nigericin stimulation, compared to neutrophils from non-infected mice (as displayed by both bioluminescence and FAM-FLICA fluorescence assays; Figs 9E and S7A, respectively). To directly assess the role of IFN-I nigericin-induced caspase1 activity, we inoculated mice with an IFN-I blocking anti-IFNAR monoclonal antibody or an isotype control antibody post-infection. Remarkably, neutrophils from anti-IFNAR treated mice showed diminished nigericin-induced caspase1 activity (Figs 9F and S7A). Furthermore, as expected due to their typical IFN-responsiveness, oasl2, caspase1 and IL-1β gene expressions were lower in anti-IFNAR treated than in isotype treated mice (Fig 9G–9I). Taken together, the results indicate that IFN-I is responsible for the increased caspase1 activity in neutrophils of infected mice.

Blocking IFN-I signaling did not significantly alter virus replication, virus-induced pathological changes, neutrophil CD11b expression or neutrophil caspase1 activity without exogenous stimuli (S7B–S7D, S7H and S7I Fig and S3 Table). Neutrophil counts indicated a significant increase in neutrophil accumulation in the lungs of anti-IFNAR treated mice; however, this was not confirmed by neutrophil/lymphocyte ratio or morphometry (S7E-S7G Fig). Interestingly, regardless of treatment, some neutrophils in infected mice displayed degeneration and NETosis, evidenced by histone H3cit staining (Fig 7J; S3 Table).

## Discussion

Neutrophils, the largest cell population of the host immune system, are rapidly recruited to sites of infection and play an important role in orchestrating an early immune response [42, 43]. The relevance of neutrophils in viral infections became increasingly apparent during the COVID-19 pandemic, as they have been shown to be key mediators of the observed pathological processes [44].

This study sheds light on the potential involvement of the inflammasome pathway in COVID-19, particularly by demonstrating its activation in mature neutrophils during SARS-CoV-2 infection. Our investigation of the inflammatory profile of neutrophils as the dominant population of peripheral blood polymorphonuclear cells (PMNs) revealed an increased ability of neutrophils from severe COVID-19 patients for inflammasome assembly as evidenced by their transcriptional profile, spontaneous release of IL-1β, and elevated caspase1 activity. These findings are consistent with previous reports indicating activation of the NLRP3 inflammasome and ASC specks in circulating neutrophils during acute COVID-19 [14, 16]. Furthermore, despite showing increased caspase1 activity, neutrophils from COVID-19 patients

exhibited diminished soluble IL-1β production upon exogenous activation of the NLRP3 inflammasome pathway compared to healthy controls, which suggests that this pathway is "exhausted" due to prior activation during the disease. Mechanistically, our findings show that IFN-I, elevated in COVID-19 patients [45, 46], can prime inflammasome formation in neutrophils. Transcriptomic analyses revealed that circulating neutrophils during severe COVID-19 show increased expression of IFN-responsive genes, suggesting inflammasome priming by IFN-I also *in vivo* during COVID-19 [47]. Furthermore, the study found that immature neutrophils, which are prevalent in low-density granulocyte fraction (LDGs), exhibit unique inflammasome gene expression and outcomes compared to mature neutrophils (PMNs). Distinctively from PMNs, LDGs do not display the IFN-I signature or upregulation of major inflammasome related genes, which indicates their lower responsiveness to IFN-I during COVID-19. However, since we were able to show that LDGs can form inflammasomes when stimulated by IFN-I *ex vivo*, their lower responsiveness is probably due to LDGs not being similarly exposed to IFN-I as compared to PMNs during COVID-19.

SARS-CoV-2 infected mice also showed increased neutrophil caspase1 activity, reversible by an IFN-I receptor (IFNAR) blocking antibody. Transcriptional analysis revealed a robust IFN-I signature and elevated expression of inflammasome genes encoding for caspase1 and IL-1β in neutrophils of infected mice, which were also inhibited by blocking IFNAR signaling, suggesting that IFN-I may also prime for inflammasome activation in mice. Notably, the anti-IFNAR treatment did not affect neutrophil recruitment or NETosis, which is consistent with another COVID-19 model using transgenic human ACE2, where IFNAR knockout inhibited recruitment of monocytes and lymphocytes, but not neutrophils, to infected lungs [48].

Inflammasomes were first studied in macrophages, revealing many molecular mechanisms regulating inflammasome assembly [49]. Macrophage inflammasome activation has emerged as a major factor also in COVID-19 [17]. Interestingly, macrophage inflammasome activation was recognized to be IFN-I mediated in an experimental rhesus macaque COVID-19 model [50]. However, due to the abundance of neutrophils compared with cells of monocyte/macrophage lineage [51,52], the significance of neutrophil inflammasomes in COVID-19 is likely underestimated. Our results highlight inflammasomes as an additional important inflammatory mechanism in neutrophils [14], complementing their role in phagocytosis, reactive oxygen species generation, degranulation, and NETosis [31].

SARS-CoV-2 can directly activate inflammasomes in cells of the monocyte/macrophage lineage [17]. Our study investigated whether SARS-CoV-2 can provide the first or second signal for inflammasome activation in neutrophils. However, we found no evidence of direct virus-induced inflammasome activation in neutrophils. The difference between macrophages and neutrophils in their susceptibility to SARS-CoV-2 could depend on many factors. Both cell types express ACE2, the receptor for SARS-CoV-2, but may differ in ACE2 expression levels [53]. Furthermore, the intracellular environment of macrophages is better suited for viral replication [54], while neutrophils focus on phagocytosis and antimicrobial responses [31,55]. Additionally, pathogen opsonization can trigger inflammasomes in macrophages [56] but is not a primary function of neutrophils. Therefore, our findings suggest neutrophil inflammasome activation in response to SARS-CoV-2 likely results from interactions with infected and/or dying cells in the lungs, rather than direct virus activation. To note, whether SARS-CoV-2 can induce neutrophil inflammasomes through immune complex-mediated mechanisms, as seen in monocytes/macrophages [17] remains to be determined.

In this study, we demonstrated IFN-I as the first signal for NLRP3 inflammasome activation in neutrophils. While prior research has explored IFN-inflammasome crosstalk [57], priming capacity of IFN-I remained unclear. While IFN-I promotes inflammasomes in epithelial cells [58] it can also dampen IL-1β in macrophages [59]. Plausibly, initial IFN-I exposure may

upregulate inflammasome genes, whereas prolonged activity could hinder IFN-I signaling via "negative feedback" loop, in line with our findings of inflammasome exhaustion in circulating neutrophils of severe COVID-19 patients. It should be noted that several SARS-CoV-2 encoded proteins have been shown to inhibit IFN-I signaling [60]. However, no evidence suggests that neutrophils can be infected by SARS-CoV-2 and therefore it seems unlikely that such direct virus mediated effects could play a role in the observed neutrophil unresponsiveness to IFN-I.

The dualistic nature of the IFN-I response in COVID-19 has been recognized previously. It seems that a strong initial IFN-I response to SARS-CoV-2 is more likely to result in asymptomatic or mild COVID-19 whereas a decreased initial IFN-I activity, due to e.g. genetic defects or increased levels of IFN-I autoantibodies, can lead to more severe COVID-19 [61]. This initial beneficial effect of IFN-I is probably due to its ability to limit viral replication at early stages of the infection. However, at later stages of the disease IFN-I can be detrimental by promoting inflammatory pathways instead of direct antiviral effects [62]. Thus, similarly to the IFN-I response in general, the role of neutrophil inflammasomes in development and severity of COVID-19 might be dualistic in nature with an initial protective effect while damaging when sustained for prolonged periods.

Our study demonstrated a strong association between PMN caspase1 activity and plasma levels of calprotectin, a marker of neutrophil activation. It is of interest to note that calprotectin can also promote inflammasome activity in neutrophils [63,64]. Therefore, in addition to IFN-I discussed in this study, it is possible that calprotectin also contributes to neutrophil inflammasome formation during COVID-19. Additionally, increased disease severity, as assessed by the WHO ordinal scale, was significantly linked to PMNs being less responsive to *ex vivo* IFN-induced inflammasome activation. Thus, these results suggest that neutrophil inflammasomes would play a role in disease severity, rather than being protective in COVID-19.

Our study also unveiled distinct gene profiles in LDGs and PMNs from severe COVID-19 patients. LDGs exhibited upregulation of genes related to DNA replication and cell cycle, indicating immaturity, and confirming our prior findings [7]. Conversely, PMNs displayed heightened NLR signaling, suggesting a robust response to pathogens. While our study compared PMNs and LDGs, and the COVID-19 Immune Atlas single cell analysis represented a broader classification of mature and immature neutrophils, the alignment of our results with the atlas provides further support for the distinct characteristics of these two neutrophil populations in severe COVID-19. Notably, IL-18 gene expression and secretion after *ex vivo* stimulation were higher in LDGs than PMNs. To note, PMN's lack of IL-18 secretion is not due to lack of protein, as they constitutively express significant amounts intracellularly [65]. This indicates a similarity between LDGs and monocytes/macrophages in inflammasome mediated IL-18 processing, possibly lost during neutrophil maturation.

The present study has some limitations worth discussing. Firstly, the relatively small human sample size may limit the generalizability of the findings. While RNA-seq provided valuable insights into gene expression profiles of PMNs and LDGs, we did not perform functional validation of the identified pathways in this study. Regarding our experimental SARS-CoV-2 disease model, the high virus input might trigger robust immune responses that differ from typical human infections, and the short-lived virus replication in the applied model does not capture the effect of prolonged antigen exposure or the complex inflammatory milieu seen in human cases. Importantly, our results do not directly assess the role of neutrophil inflammasomes in COVID-19 pathogenesis in humans or in the animal disease model. Further studies are therefore needed to understand the relative contribution of neutrophil inflammasomes in COVID-19 disease progression, compared to the better described macrophage inflammasomes

as well as to other inflammatory pathways engaged by neutrophils such as degranulation, reactive oxygen species production and NETosis. Furthermore, due to the ubiquitous expression of IFNAR, the observed inhibitory effects on neutrophil inflammasome activity by IFNAR blockade does not exclude the possibility that IFN-I could promote neutrophil inflammasome formation by indirect effects such as regulating the interplay between neutrophils and other immune cells or stimulating the release of pro-inflammatory cytokines by other cell types. In addition, investigating the effects of IFNAR blockade in other time points than the chosen 2 dpi might have been more valuable in revealing its effects on viral replication and neutrophil accumulation in the infected lungs. Finally, since the prominent role of neutrophils in the immune response to viral infections is widely recognized [42,43] and it would be valuable to compare these findings to neutrophil responses in other viral respiratory infections.

Taken together, our findings provide valuable insights into neutrophil involvement in COVID-19 and possibly other viral respiratory infections. However, further research is needed to fully grasp the role of neutrophil inflammasomes in COVID-19 pathogenesis. This increased understanding may facilitate the development of targeted treatment approaches for COVID-19. For example, pharmacologically targeting the inflammasome pathway in neutrophils with novel inhibiting molecules [66], may help mitigate the exaggerated inflammatory response observed in severe cases. The next steps involve validating the pathways and genes identified as potential therapeutic targets and assessing their COVID-19 specificity. Prospectively, these strategies could be extended to address upcoming respiratory virus pandemics, where neutrophils and inflammasomes provide major pathogenic contributions.

## Supporting information

**S1 Fig. Comparison of gene expression in granulocyte populations of COVID-19 patients using RNA-seq analysis.** (**A**) Deconvoluted RNA-seq data. The cellular composition in isolated PMN and LDG fractions was estimated using CIBERSORTx through the identification of cell populations based on RNA-seq. The bar plots in the figure represent the cell composition of each RNA-seq sample, offering insights on sample purity. (**B**) Heatmap of the top 118 differentially expressed genes between PMNs from healthy controls, mild and severe COVID-19, as well as LDGs from severe disease, identified by unsupervised ICGS analysis based on correlation, using AltAnalyze software. IFN-related genes, identified by GENESHOT, are shown in bold.
(TIFF)

**S2 Fig. Enriched differentially expressed genes and pathways in severe COVID-19 PMNs and LDGs.** (**A-B**) Volcano plots of DEGs between severe COVID-19 PMNs versus (**A**) severe COVID-19 LDGs and (**B**) HC PMNs. (**C-D**) Volcano plots of enriched gene sets in severe COVID-19 PMNs versus (**C**) HC PMNs and (**D**) severe COVID-19 LDGs, using KEGG database. Each point represents a single gene set, where the x-axis measures its odds ratio, while the y-axis shows its -log10(p-value). (**E-F**) Volcano plots of (**E**) severe COVID-19 PMNs versus mild COVID-19 PMNs and (**F**) mild COVID-19 PMNs vs HC PMN. For all panels, blue points represent significant terms (adjusted p-value < 0.05), while smaller gray points represent non-significant terms. *DEG = differentially expressed genes.*
(TIFF)

**S3 Fig. Differential expression of interferon and inflammasome related genes in PMNs during COVID-19.** RNA was extracted from isolated HC PMNs (n = 8–13) versus severe COVID-19 PMNs (n = 29–32) and subjected to comparative RT-qPCR using specific primers for OAS1, OAS2, IFIT1, IFI16, caspase1, caspase5, IL1B, NLRC4, NLRC5, NLRP3 and NAIP.

*p < 0.05, **p < 0.01, ***p < 0.001 and **** p < 0.0001. P values calculated with Mann-Whitney U-test. Data presented as mean ± SD.
(TIFF)

**S4 Fig. Expression of inflammasome related genes in mature and immature neutrophils from COVID-19 PBMCs.** The fraction of mature and immature neutrophils cells expressing 17 inflammasome related genes identified in Fig 2D (shown in black and blue, respectively) are shown in a bar graph. For each gene, the proportion of expressing cells is shown in light blue, while the proportion of negative or not-expressing cells is shown in gray. Zoomed-in bar graph depicts the proportion of mature and immature cells expressing each gene.
(TIFF)

**S5 Fig. *Ex vivo* stimulation of isolated PMNs.** (**A**) HC PMNs (1 million/ml) were primed for 4 hr by low dose IFN-α, low dose IFN-β (both $2.7*10^3$ IU/ml) or 20 ng/ml LPS followed by 2.5 μM nigericin activation for 4 hr. IL-1β release was measured by ELISA and the assays were performed in the presence of either α-IFNAR1 or mouse IgG as control (both 100 μg/ml) (n = 5). *p < 0.05 and **p < 0.01. P values calculated using two-way ANOVA with Šídák multiple comparison test. (**B**) IL-18 (n = 2–3 HC PMN and 3 COVID-19 PMN) was measured from supernatants by ELISA following LPS or IFN-I priming (4 h) and subsequent nigericin activation (4 h). (**C-E**) Effect of different inflammasome specific inhibitors in cytokine secretion. (**C**) Effect of inflammasome inhibitor MCC950 (2 μg/ml) and YVAD (20 μg/ml) on LPS or IFN-I primed (4 h) and nigericin activated (4 h) IL-1β secretion in the supernatant of healthy control PMNs (n = 8). (**D-E**) Effect of inflammasome inhibitor MCC950 (2 μg/ml, added simultaneously with nigericin) on LPS or IFN-I primed (4 h) and nigericin activated (20 h). *p < 0.05, **p < 0.01, ***p < 0.001, **** p < 0.0001. P values calculated with Kruskal-Wallis test. Data presented as mean ± SD. *IFN = interferon type I, LPS = lipopolysaccharide, Nig = nigericin, YVAD = tetrapeptide caspase1 inhibitor Tyr-Val-Ala-Asp*. (**F-I**) Gene expressions in HC and COVID-19 PMNs after LPS or IFN-I stimulation. A comparison of gene expression in isolated healthy control PMNs versus COVID-19 PMNs after *ex vivo* stimulation with LPS or IFN-I. Extracted RNA was subjected to comparative RT-qPCR using specific primers for NLRP3, NLRC4, NAIP and CASP5 (n = 4–8 for HC PMN and 6–9 for COVID-19 PMN). *p < 0.05. Two-way ANOVA with Tukey's multiple comparison test was applied. Data were presented as mean ± SD.
(TIFF)

**S6 Fig. Animal weight dynamics, flow cytometry gating strategy and immunohistochemistry in SARS-CoV-2 infected mice.** Female BALB/c mice were intranasally inoculated with $5*10^5$ TCID50 SARS-CoV-2 MaVie strain or PBS as control and euthanized at 2 dpi or 4 dpi. (**A**) Daily tracking of animal weight performed throughout the experiment (n = 12 for SARS-CoV-2 infected animals, n = 6 for PBS-inoculated animals). The weights of the mice euthanized at 2 dpi (n = 26) did not show significant differences and are not reported. (**B**) Gating strategy to analyze Ly6G+ neutrophils in mouse lung single cell suspensions. Side scatter area (SSC-A) versus forward scatter area (FSC-A) plot followed by side scatter area versus height (SSC-A vs SSC-H) plot were used for the identification of single cells. BV605 yellow live/dead dye was used to discriminate dead cells, from which CD3/CD19+ lymphocytes and Ly6G + neutrophils were gated as shown. (**C**) Representative histogram showing the percentage of Ly6G+ cells after isolation from lung single cell suspension using Ly6G-binding magnetic beads. (**D**) Left column: immunohistochemistry for SARS-CoV-2 nucleoprotein; right column: immunohistochemistry for Ly6G (neutrophil marker), hematoxylin counterstain.
Bars = 500 μm (large images) and 50 μm (insets). At 2 dpi (top), the arrow points at a bronchus

with viral antigen expression in epithelial cells. A close–up of the bronchus (bottom; B: bronchial lumen) shows degenerated and slough off antigen positive epithelial cells. Adjacent alveoli exhibit viral antigen expression in typeI (arrowhead) and typeII (arrow) pneumocytes. The overview (top) shows neutrophils between the infected bronchial (arrow) epithelial cells, in parenchymal areas (arrowhead; right inset) and in capillaries (arrowheads). A close-up of the bronchus (bottom; B: bronchial lumen) highlights numerous neutrophils between degenerate (arrowheads) epithelial cells. At 4 dpi (middle), there are focal areas with antigen expression in alveolar epithelial cells and infiltrating macrophages. Neutrophils are present among the infiltrating cells (arrow) as individual cells (inset: arrows) or in aggregates (inset: arrowhead). The bottom shows the lung of a mock-infected control animal. There is no viral antigen expression. Staining for Ly6G depicts individual neutrophils in larger vessels (inset: arrow) or in capillaries (inset: arrowheads).
(TIFF)

**S7 Fig. Impact of α-IFNAR treatment in SARS-CoV-2 infected mice.** Female BALB/c mice were intranasally inoculated with $5* 10^5$ TCID50 SARS-CoV-2 MaVie strain or PBS as control. Mice were intraperitoneally inoculated with 250 μg anti-IFNAR or IgG1 isotype control directly after infection with SARS-CoV-2 and lung neutrophils isolated at 2 dpi (including also intranasally PBS-inoculated control mice without intraperitoneal injection) (**A**) Quantification of caspase1 positive cells in nigericin-activated isolated Ly6G neutrophils stained by FAM-FLICA. Representative histogram is shown. * $p < 0.05$. P values calculated using one-way ANOVA with multiple comparison test (Holm-Šídák correction). (**B**) RNA was isolated from mouse lungs and subjected to RT-qPCR targeting the replication-intermediate subgenomic E gene and GAPDH as housekeeping gene. RNA levels were assessed based on cycle threshold Ct levels. The expression levels of the target gene SubE were measured and normalized to GAPDH levels using the comparative Ct method (ΔΔCt). The fold change values were calculated by the formula $2^{(-\Delta\Delta Ct)}$, representing the relative gene expression compared to the PBS mock-infected control (in which subE was undetectable but set to 40 Ct). No significant differences are seen between the two groups, assessed with Welch's t-test. (**C**) Infectious virus was calculated from supernatants of lung single cell suspensions of infected mice as fluorescence focus forming units (FFU) in Vero E6 cells. (**D**) Quantification based on morphometric analysis that determines the area of immunolabelling for SARS-CoV-2 nucleoprotein in relation to total tissue area. (**E**) Quantification of Ly6G neutrophil/lymphocyte ratio in lung single cell suspensions by flow cytometry. (**F**) Quantification of Ly6G cell counts extrapolated per lung in single cell suspensions by flow cytometry (**G**) Quantification of Ly-6G based on morphometric analysis that determines the area of immunolabelling for Ly6G in relation to total tissue area in mock-infected controls. (**H**) Quantification of median fluorescence intensity (MFI) of CD11b expression in ly6G neutrophils by flow cytometry. (**I**) Quantification of FAM-FLICA MFI in Ly6G+ neutrophils by flow cytometry. Panels B, D and G are representative of two independent experiments. (**J**) Histological features, viral antigen expression and extent of neutrophil influx and evidence of neutrophil damage in the lung of SARS-CoV-2 infected BALB/C mice after isotype control and anti-IFNAR treatment at 2dpi. Left column: Control isotype treated mice; right column: anti-IFNAR treated mice. HE stain (top layer) and immunohistology, hematoxylin counterstain (all other images). Bars: 250 μm (overview images) and 25 μm (insets). In control isotype treated mice, the lung exhibits degeneration and loss of bronchial and bronchiolar epithelial cells (HE stain: arrowhead; right inset), with mild inflammatory infiltration. The parenchyma exhibits focal areas of increased cellularity, with typeII pneumocyte activation and occasional degenerate alveolar epithelial cells (arrows; left inset: degenerate cells (arrowhead) and infiltrating neutrophil (arrow)). Staining for

SARS-CoV-2 NP confirms epithelial cell infection in bronchus (arrowhead; right inset) and alveoli (arrow; left inset). Right inset: Viral antigen expression is seen in intact and sloughed off, degenerate epithelial cells. Left inset: Viral antigen expression is seen in both typeI (small arrowhead) and typeII (small arrow) pneumocytes; there are also degenerate positive cells (large arrowhead). Neutrophils (Ly6G+) are located within focal parenchymal areas of increased cellularity (arrows; left inset: arrowheads) and present between degenerate bronchial epithelial cells (arrowhead; right inset: arrowhead). Staining for histone H3 shows neutrophil degeneration/NETosis in parenchymal areas (arrow; left inset: arrowheads) and associated with degenerate epithelial cells (arrowhead; right inset: positive reaction between sloughed off epithelial cells (arrow) and between the intact epithelial layer (arrowhead)). In anti-IFNAR treated animals, the lung exhibits degeneration and loss of bronchial and bronchiolar epithelial cells (arrowhead; right inset: arrows), with mild inflammatory infiltration and individual neutrophils between intact and sloughed off degenerate epithelial cells (right inset: arrowheads). The parenchyma exhibits focal areas of increased cellularity, with typeII pneumocyte activation and occasional degenerate alveolar epithelial cells (arrows; left inset: degenerate cells (arrow) and infiltrating neutrophils (arrowhead). Staining for SARS-CoV-2 NP shows epithelial cell infection in bronchioles (arrowhead; right inset) and alveoli (arrow; left inset). Right inset: Viral antigen expression is seen in intact and sloughed off, degenerate epithelial cells. Left inset: Viral antigen expression is seen in pneumocytes (arrow) and infiltrating macrophages (arrowheads). Neutrophils (Ly6G+) locate within focal parenchymal areas of increased cellularity (arrows; left inset) and are present between intact (inset: arrowhead) and degenerate epithelial cells (arrowhead; right inset: arrow). Staining for histone H3cit shows neutrophil degeneration/NETosis in parenchymal areas (arrow; left inset) and associated with degenerate epithelial cells (arrowhead; right inset: positive reaction between sloughed off epithelial cells (arrow) and between the intact epithelial layer (arrowhead)). *Dpi = days post infection; NP = nucleoprotein.*
(TIFF)

**S1 Table. Information for Fig 3, detailing the Gene Set Enrichment Analysis (GSEA) databases used for pathway analyses.**
(XLSX)

**S2 Table. qPCR primer sequences: gene-specific forward and reverse primers.**
(DOCX)

**S3 Table. Histological changes as well as SARS-CoV-2 nucleoprotein and RNA expression in female BALB/C mice infected with SARS-CoV-2.**
(DOCX)

**S1 Source Data. Original scan of IL-1β and actin western blot. HC PMNs were non-stimulated or stimulated 4h with IFN-I (combination of 2.7\*104 IU/ml IFN-α and IFN-β) or 20 ng/ml LPS (1st signal), followed by 4h with 2.5 μM nigericin (2nd signal).** Western blots from supernatants and cell lysates were performed first for actin and then for IL-1β on the same membrane as indicated. Information for Fig 5A.
(PDF)

## Acknowledgments

RNA isolation, library preparations and RNA sequencing was performed at the Institute for Molecular Medicine Finland FIMM, Genomics unit supported by HiLIFE and Biocenter Finland. The authors also thank M. Utriainen for expert technical assistance.

## Author Contributions

**Conceptualization:** Luz E. Cabrera, Tomas Strandin.

**Data curation:** Luz E. Cabrera, Tomas Strandin.

**Formal analysis:** Luz E. Cabrera, Tomas Strandin.

**Funding acquisition:** Anu Kantele, Olli Vapalahti, Tomas Strandin.

**Investigation:** Luz E. Cabrera, Suvi T. Jokiranta, Sanna Mäki, Simo Miettinen, Lauri Kareinen, Anja Kipar, Tomas Strandin.

**Methodology:** Anja Kipar, Tomas Strandin.

**Project administration:** Tomas Strandin.

**Resources:** Ravi Kant, Tarja Sironen, Jukka-Pekka Pietilä, Anu Kantele, Hanna Lindgren, Pirkko Mattila, Olli Vapalahti.

**Software:** Luz E. Cabrera.

**Supervision:** Eliisa Kekäläinen, Anja Kipar, Tomas Strandin.

**Validation:** Luz E. Cabrera, Anja Kipar, Tomas Strandin.

**Visualization:** Luz E. Cabrera, Anja Kipar.

**Writing – original draft:** Luz E. Cabrera, Tomas Strandin.

**Writing – review & editing:** Suvi T. Jokiranta, Lauri Kareinen, Anu Kantele, Eliisa Kekäläinen, Pirkko Mattila, Anja Kipar.

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
