## [Decision Letter · Decision Letter 0]

26 Feb 2024

Dear Ms Cabrera,

Thank you very much for submitting your manuscript "The assembly of neutrophil inflammasomes during COVID-19 is mediated by type I interferons" for consideration at PLOS Pathogens. As with all papers reviewed by the journal, your manuscript was reviewed by members of the editorial board and by several independent reviewers. In light of the reviews (below this email), we would like to invite the resubmission of a significantly-revised version that takes into account the reviewers' comments.

Two additional expert reviewers have provided comments. They both view your report to be important and with potential to inform new therapies for COVID19. Patient samples and a focus on understudied cell types influencing COVD19 make this submission valuable to the field. However, the reviewers expected additional results that are necessary to identify the relative contribution of neutrophil inflammasomes to severe COVID19. First, both reviewers noted that flow cytometry can reveal multiple signatures of isolated PMNs, without requiring ex vivo stimulation. Such flow results can help explain how PMNs may operate in COVID19. Second, reviewer 2 recommended orthogonal approaches to measure caspase activation, as the results may address the spontaneous IL1beta releases and other discrepancies in the current data. Both of these experiments should be performed and the new results presented in a revised submission. Note that many of the minor issues may be suitably addressed with modifications to the text.

We cannot make any decision about publication until we have seen the revised manuscript and your response to the reviewers' comments. Your revised manuscript is also likely to be sent to reviewers for further evaluation.

Sincerely,

Tom Gallagher

Guest Editor

PLOS Pathogens

Alexander Gorbalenya

Section Editor

PLOS Pathogens

Michael Malim

Editor-in-Chief

PLOS Pathogens

orcid.org/0000-0002-7699-2064

Editor comments:

Dear Dr. Cabrera, two additional expert reviewers have provided comments. They both view your report to be important and with potential to inform new therapies for COVID19. Patient samples and a focus on understudied cell types influencing COVD19 make this submission valuable to the field. However, the reviewers expected additional results that are necessary to identify the relative contribution of neutrophil inflammasomes to severe COVID19. First, both reviewers noted that flow cytometry can reveal multiple signatures of isolated PMNs, without requiring ex vivo stimulation. Such flow results can help explain how PMNs may operate in COVID19. Second, reviewer 2 recommended orthogonal approaches to measure caspase activation, as the results may address the spontaneous IL1beta releases and other discrepancies in the current data. Both of these experiments should be performed and the new results presented in a revised submission. Note that many of the minor issues may be suitably addressed with modifications to the text.

Reviewer's Responses to Questions

**Part I - Summary**

Reviewer #1: In their current revision, the authors made modifications according to previous review. The manuscript is improved in clarifying their major information. However, current data and discussion fail to offer the direct answer to the question raised by previous review: "what is the contribution of neutrophil inflammasomes to the severity of COVID-19". Although PMN isolated from severe cases show different characteristics in ex vivo experiments compared to those isolated from healthy donors, it is not hard evidence supporting their in vivo role. In addition, as pointed out by previous review, no clear effects of PMN inflammasome were found in animal model which further ruin the persuasion of authors' conclusion.

Unfortunately, the authors declined the application of flow cytometry in their additional experiment. In addition to its quantitative function, flow cytometry can also show the expression of multiple molecular signatures of isolated PMN without further ex vivo stimulation, which may represent the real situation of PMN in severe COVID-19 cases and their healthy controls.

Finally, the authors refined their discussion on the different roles of PMN and LDG. However, the LDG data is still ambiguous and doesn't help clarify the role of PMN inflammasome in severe COVID-19. I would recommend removing them at all.

Reviewer #2: Overall, this is an exciting paper that provides elegant RNA sequencing data from patient samples with ex vivo assays and an in vivo mouse infection model that further characterize potential mechanisms of inflammasome activation in neutrophils in response to SARS-CoV-2 infection/COVID-19. Although NLRP3 inflammasome activation has been demonstrated in neutrophils previously, the specific interrogation of inflammasome pathways in low-density granulocyte (LDGs) versus mature neutrophils (PMNs) and mechanistic involvement of IFN signaling in this pathway is novel. Importantly, the scope of this work extends beyond COVID-19, as similar pathways are likely to be activated in neutrophils in other viral infection models or inflammatory diseases.

The paper would benefit from additional assays to confirm caspase-1 activation (such as ASC-speck formation, FLICA, or WB for caspase-1 or Gasdermin-D cleavage) in mouse neutrophils isolated from infected lungs and in COVID-19 PMNs, which may help resolve discrepancy between the spontaneous IL-1b release measured (fig. 2C) despite no detectable caspase-1 activation (as measured by luciferase) in the absence of stimulation (fig. 2G). Other experiments suggested, including more robust phenotypic characterization of the neutrophils recruited to the mouse lungs in the in vivo Sars-CoV-2 infection model by flow cytometry would also improve the manuscript.

**Part II – Major Issues: Key Experiments Required for Acceptance**

Reviewer #1: See above in Summary.

Reviewer #2: Fig 2C shows spontaneous IL-1b release (i.e. no signal 2 required) in COVID-19 PMNs but no difference in caspase-1 activation in Fig 2G - only see casp-1 activation in response to nigericin. But IL-1b release is presumably caspase-1 dependent. Can the authors measure caspase-1 activation by another approach? Such as staining for ASC-specks, FLICA, or WB (looking at caspase-1 or gasdermin D cleavage)? Any additional assay confirming caspase-1 activity in COVID-19 PMNs would increase rigor and add significantly to the paper

Fig 4B seems to conflict with Fig 2C and Fig 2G, which shows COVID-19 PMNs display spontaneous IL-1b release and casp-1 activation in response to nigericin, respectively. In Fig 4B, COVID-19 PMNs (mock or nigericin only treated) don’t exhibit any increased IL-1b release. This discrepancy should be discussed

The authors state that “neutrophils may lose the ability to secrete IL-18 during maturation” (lines 511-512). Can the authors directly compare cytokine production between HC PMNs, COVID-19 PMNs and COVID-19 LDGs (+/- stimulation)? These data would better support the conclusion drawn, as the following discrepancies contradict this conclusion: In Fig 2, there is no difference in IL-18 secretion between HC and COVID-19 PMNs, but the raw values are ~50pg/mL. In Supplement 4D, there is no difference between stimulated and unstimulated HC PMNs, raw values ~40 pg/mL. In Fig 6, the raw values of IL-18 release from LPS+nig-stimulated COVID-19 LDGs is also ~50 pg/mL, but appear higher relative to untreated LDGs, which exhibit little-to-no cytokine production. Taken together, it doesn't seem like neutrophils are losing their ability to secrete IL-18 as they differentiate. Plos does not accept references to data not shown (line 511), but if the authors show these data it would be a good demonstration to support the above conclusion that inflammasome-dependent cytokine release in neutrophils is dependent on their differentiation status

The in vivo experiments using the mouse Sars-CoV-2 infection model are critical to validate the author's RNA seq analyses and ex vivo characterization of neutrophil populations in COVID-19, and therefore, Fig 7 would benefit from additional data. Can the authors perform flow cytometry on cells in the lung to 1. validate neutrophil recruitment during infection (quantification of Ly6G-pos cells), 2. phenotype these cells (are the Ly6G-pos cells mature or immature neutrophils?) and 3. quantify caspase-1 activation in lung neutrophils during infection (using a caspase-1 Flica-probe or by measuring ASC specks in isolated neutrophils). As an alternative approach, is inflammasome-dependent cytokines upregulated in the BALF 2 dpi (when peak neutrophil recruitment is measured) and is cytokine production inhibited by the anti-IFNAR treatment?

**Part III – Minor Issues: Editorial and Data Presentation Modifications**

Reviewer #1: See above in Summary.

Reviewer #2: In lines 409 – 410, the authors state that the non-COVID-19 patients presented with ARDS and clinical concern for COVID-19 but tested negative for SARS-CoV2 by PCR. It is not clear if the ‘non-COVID-19 patients were once seropositive and this distinction is important for the conclusion drawn. Perhaps more accurate terminology would be Sars-CoV-2 PCR positive vs. PCR negative?

Could the authors provide a list of genes for the pathways shown in Figure 3, provided as a supplement?

The authors conclude that Sars-CoV-2 particles have no (direct) effect on PMN inflammasome activation, but viral particles are only used as 'signal 2' in these experiments and IL-1b release appears to be nigericin dependent. Did the authors test if viral ligands instead prime PMNs for inflammasome activation? Is IL-1b released from HC PMNs stimulated with Sars-CoV-2 particles + nigericin?

Could the authors further investigate the relationship between calprotectin and inflammasome activation in Fig. 5?

1. Is S100A8 and A9 upregulated in the RNAseq data set in severe COVID-19 PMNs?

2. Since calprotectin levels are elevated in COVID-19 patients, it would be interesting to see if calprotectin can directly activate the inflammasome in neutrophils. Can the authors treat PMNs with S100A8/A9 (+/-nigericin) and measure caspase-1 activation or cytokine production?

3. Alternatively, does inflammasome activation promote S100A8/A9 release from neutrophils? This was recently demonstrated in PMID 37903858. It would be interesting to see if ex vivo LPS/IFN + nig stimulation of HC PMNs drives S100 production

Fig 5C legend should be IFN + nig (not LPS)

Supplementary Fig 5 (lines 492-493) – ASC transcript expression does not confirm inflammasome forming capacity of cells; inflammasomes can form in the absence of ASC

Line 502 - Wrong fig reference? Should be D-E

In Supplement 6A and 7B, can the authors quantify viral load by PFU or genome copy number by qRT-PCR?

In Supplement 6B, viral antigen quantification should also be shown for PBS (as done in panel C).

The authors conclude that their findings “suggest a pivotal role of neutrophils in clearing the virus in SARS-CoV-2 infected mice” (lines 534-535), when viral clearance was not measured. The data demonstrate neutrophil recruitment to the lungs in this mouse model, not a pivotal role for neutrophils in clearing viral infection.

Supplemental data depicting purity of isolated neutrophils (by flow cytometry) for Fig 7 should be included

In Supplement 7C, representative H3cit IHC staining should be shown for PBS/mock-infected mice; quantifications should also be shown, as in Supplement 6

PLOS authors have the option to publish the peer review history of their article (what does this mean?). If published, this will include your full peer review and any attached files.

Reviewer #1: No

Reviewer #2: No

Figure Files:

Data Requirements:

Reproducibility:

To enhance the reproducibility of your results, we recommend that you dep

---

## [Decision Letter · Decision Letter 1]

26 May 2024

Dear Ms Cabrera,

Thank you very much for submitting your manuscript "The assembly of neutrophil inflammasomes during COVID-19 is mediated by type I interferons" for consideration at PLOS Pathogens. As with all papers reviewed by the journal, your manuscript was reviewed by members of the editorial board and by several independent reviewers. The reviewers appreciated the attention to an important topic. Based on the reviews, we are likely to accept this manuscript for publication, providing that you modify the manuscript according to the review recommendations.

Note that reviewers are generally satisfied with your improved submission. One reviewer suggested additional efforts to reorganize text and emphasize the main theme of the works. If results addressing specific comments from reviewer 1 are available, then please add these to the manuscript. If not available, then please address the comments with text additions that indicate the limitations of the current data and the importance of future experiments that may provide greater confidence in the conclusions.

Sincerely,

Tom Gallagher

Guest Editor

PLOS Pathogens

Alexander Gorbalenya

Section Editor

PLOS Pathogens

Michael Malim

Editor-in-Chief

PLOS Pathogens

orcid.org/0000-0002-7699-2064

Editors comments:

Note that reviewers are generally satisfied with your improved submission. One reviewer suggested additional efforts to reorganize text and emphasize the main theme of the works. If results addressing specific comments from reviewer 1 are available, then please add these to the manuscript. If not available, then please address the comments with text additions that indicate the limitations of the current data and the importance of future experiments that may provide greater confidence in the conclusions.

Reviewer Comments (if any, and for reference):

Reviewer's Responses to Questions

**Part I - Summary**

Reviewer #1: Cabrera et al's manuscript "The assembly of neutrophil inflammasomes during COVID-1 19 is mediated by type I interferons" demonstrates the inflammatory characteristics of neutrophils in COVID-19 patients and SARS-CoV-2-infected mice. The study is interesting and consists of an impressive amount of data. However, the paper needs to be further organized to highlight the major findings. Currently, the authors describe the changes of PMN, LDG, inflammasome, and IFN-I by using both human COVID-19 patient samples and mice model, but it is hard to identify the main theme of the story.

Reviewer #2: All comments were adequately addressed. New experiments added demonstrating caspase-1 activation in lung neutrophils isolated from infected mice add significantly to the paper.

**Part II – Major Issues: Key Experiments Required for Acceptance**

Reviewer #1: 1. I agree with Reviewer 1 that "no clear effects of PMN inflammasome were found in animal model". In vivo depletion or blockade assay is needed.

2. I also agree with Reviewer 1 on "the authors refined their discussion on the different roles of PMN and LDG. However, the LDG data is still ambiguous and doesn't help clarify the role of PMN inflammasome in severe COVID-19". If removal of these data does not apply, the better organization is necessary.

3. In addition, if the authors' explanation (increased IL-18 production in LDG results from its constituent expression) is correct, then type I IFN is dispensable for the universal inflammasome expression of neutrophils (including PMN and LDG), which impair the conclusion of this manuscript.

4. In addition to flow cytometry, identifying the phenotype of neutrophils in lung slides by confocal microscopy may help illustrate their potentially pathogenic roles during SARS-CoV-2 infection.

5. Finally, the blockade of IFNAR is not cell-specific. Therefore, the blocking effects on other immune cells, either quantity or quality, need to be addressed.

Reviewer #2: None

**Part III – Minor Issues: Editorial and Data Presentation Modifications**

Reviewer #1: N/A

Reviewer #2: None

PLOS authors have the option to publish the peer review history of their article (what does this mean?). If published, this will include your full peer review and any attached files.

Reviewer #1: No

Reviewer #2: No

Figure Files:

Data Requirements:

Reproducibility:

References:

---

## [Editor Report · Decision Letter 2]

24 Jun 2024

Dear Ms Cabrera,

We are pleased to inform you that your manuscript 'The assembly of neutrophil inflammasomes during COVID-19 is mediated by type I interferons' has been provisionally accepted for publication in PLOS Pathogens.

Best regards,

Tom Gallagher

Guest Editor

PLOS Pathogens

Alexander Gorbalenya

Section Editor

PLOS Pathogens

Michael Malim

Editor-in-Chief

PLOS Pathogens

orcid.org/0000-0002-7699-2064
---

## [Editor Report · Acceptance letter]

31 Jul 2024

Dear Ms Cabrera,

We are delighted to inform you that your manuscript, "The assembly of neutrophil inflammasomes during COVID-19 is mediated by type I interferons," has been formally accepted for publication in PLOS Pathogens.

Best regards,

Michael Malim

Editor-in-Chief

PLOS Pathogens

orcid.org/0000-0002-7699-2064